# Cleaning label noise with vision-language models

## Abstract

Current mainstream methods for learning with noisy labels often rely on sample selection, such as the common 'small-loss' strategy that considers samples with smaller losses as clean. Following this, most research focuses on developing more robust sample selection strategies. However, they are still influenced by problems such as the 'self-confirmation bias', which stems from their reliance on the in-training model. Furthermore, relying solely on visual information for sample selection can introduce biases and challenges, such as the common issue of 'hard noise', where samples are erroneously labeled as semantically similar categories. To address these challenges, this paper proposes using the popular vision-language model CLIP for sample selection. Leveraging CLIP, a pre-trained model, can effectively mitigate self-confirmation bias. Additionally, CLIP's distinctive language modality supplements potential biases introduced by relying solely on visual information for sample selection. Specifically, we introduce the *CLIPSelector*, which utilizes both the CLIP's zero-shot classifier and an easily-inducible classifier based on its vision encoder and noisy labels for sample selection. We theoretically and empirically demonstrate the unique advantages of the *CLIPSelector*. To evaluate its effectiveness on existing benchmarks, we further introduce a semi-supervised learning method called *MixFix*, tailored for noisy datasets. *MixFix* leverages the subset selected by the *CLIPSelector* and gradually introduces missing clean samples and re-labeled noisy samples based on different thresholds. In comparison to current hybrid methods involving iterative sample selection and multiple off-the-shelf techniques like model co-training, our approach simplifies the process. Nonetheless, our approach achieves competitive or superior performance across various benchmarks, including datasets with synthetic and real-world noise. Code will be released upon acceptance.

## 1 Introduction

Over the past two decades, deep neural networks have demonstrated exceptional success in various vision tasks, attributed to the existence of high-precision, large-scale datasets such as ImageNet-1K. However, collecting high-quality labels for such datasets is generally a time-consuming and labor-intensive process. To mitigate the cost, an alternative is automatic labeling (e.g. "webly-labeled" dataset by web-crawling the images and labels). While reducing the time and cost of manual labeling, it inevitably leads to low-quality noisy labels.

To address the problem of label noise, a variety of methods have been proposed. Some methods, aim to develop robust loss functions (Zhang & Sabuncu, 2018; Ghosh et al., 2017; Wang et al., 2019) or noise transition matrix (Goldberger & Ben-Reuven, 2016; Patrini et al., 2017; Hendrycks et al., 2018). However, in practice, these methods are often sub-optimal dealing with high noise ratio and complicated noise. More recently, methods based on sample selection (Sun et al., 2022; Wei et al., 2022; Wang et al., 2022; Karim et al., 2022; Patel & Sastry, 2023; Zhang et al., 2021a) to filter out samples with noisy labels become perhaps the dominant paradigm. For example, the most common sample selection strategy is the 'small-loss' mechanism motivated by the memorization effect (Arpit et al., 2017), that is, the model tends to fit clean samples earlier than noisy samples in the training process thus resulting in relatively smaller losses for the clean ones. Following this, most of methods focus primarily on improving sample selection mechanisms, including different variants of 'small-loss' strategy (Li et al., 2020a; Xia et al., 2021; Arazo et al., 2019), and utilizing kNN (Bahri

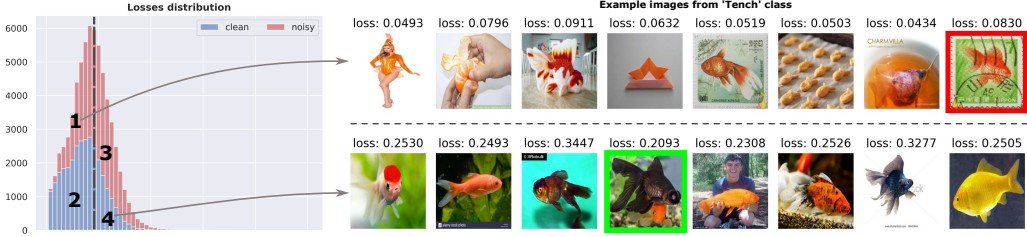

Figure 1: The losses distribution of WebVision dataset after one epoch warmup training, i.e., training with whole dataset and cross-entropy loss. Here 'clean'/'noisy' denotes samples been identified as clean/noisy by *CLIPSelector* while the 'gray vertical line' denotes the sample selection boundary induced by 'small-loss' mechanism. We show some example images on part **1** and part **4** which represents samples identified as 'clean' by 'small-loss' while rejected by *CLIPSelector* and vice versa. Especially, we mark two specific samples[1] from the semantic class 'Tench' with red and green. The red one is a post stamp of the tench fish which is very semantically similar to the real tench images thus with smaller loss. While the green one is actually a photo of pop-eyed goldfish however with black color which is more different than the common tench fish with golden color.

et al., 2020; Ortego et al., 2021; Feng et al., 2022) or graph models (Wu et al., 2020; 2021) based on samples' feature space for sample selection. However, these methods are inherently affected by the label noise as they still rely on the current in-training model, leading to the infamous self-confirmation bias. Some methods (Han et al., 2018; Yu et al., 2019) attempt to alleviate self-confirmation bias through model co-training, but this approach noticeably introduces additional computational overhead. Moreover, these methods solely rely on the visual information within the images, which can readily lead to biased sample selection outcomes, as exemplified in cases of 'hard noise' - noisy sample exhibits a highly visual similarity with incorrectly labeled classes, as illustrated in fig. 1.

To address the aforementioned issues, this paper proposes utilizing popular vision-language model - CLIP (Radford et al., 2021) for sample selection. As a pre-trained foundation model, CLIP is unaffected by the label noise in the collected dataset thus avoiding self-confirmation bias. More importantly, CLIP's distinctive language modality and zero-shot classifier allow us to compensate for the biases that may arise from solely relying on visual information for sample selection. For instance, this allows us to identify 'hard noise' (fig. 1) that is difficult to distinguish using only the visual modality. To the best of our knowledge, we are the first to employ a large-scale vision-language model, particularly leveraging its language modality, for sample selection. Specifically, we simultaneously utilize CLIP's zero-shot classifier and an easily-inducible classifier based on noisy labels and CLIP's vision encoder. We name this method *CLIPSelector* and theoretically and empirically demonstrate its effectiveness and unique advantages. Furthermore, to evaluate the performance of *CLIPSelector* on existing datasets, we introduce a straightforward semi-supervised learning method tailored for noisy datasets, namely *MixFix*. In detail, based on the subset selected by *CLIPSelector* we gradually introduce ignored clean samples and re-labeled noisy samples into the initial subset using two different thresholds and then perform class balancing to obtain the final training set.

By leveraging *CLIPSelector* and *MixFix* we establish a simple two-step framework to facilitate learning in the presence of noisy labels: we initiate with sample selection for noisy datasets using *CLIPSelector* and then perform pure semi-supervised learning using *MixFix*. Compared to existing methods involving multiple iterations of sample selection and model training, our approach features a simpler structure and aligns better with end-to-end training logic when the noise information in the dataset is agnostic. Despite its simplicity, our method achieves competitive and superior performance on various datasets, including CIFAR10/CIFAR100 with synthetic noise (symmetric, asymmetric, and instance-dependent noise), as well as real-world noisy datasets like Red Mini-ImageNet, WebVision, Clothing1M, and ANIMAL-10N.

---

[1]We can not find the specific source of red image, but highly-related images can be found with keyword: `1966 Japanese Goldfish Stamp Postage`, while the green one can be originated back to `https://acnl.fandom.com/wiki/Pop-Eyed_Goldfish`.

## 2 RELATED WORKS

**Sample selection for noisy dataset**  Most of the recent sample selection methods do so, by relying on the in-training model, for example the per-sample losses (Arazo et al., 2019; Li et al., 2020a; Han et al., 2018; Yu et al., 2019; Jiang et al., 2018) or model predictions (Song et al., 2019; Malach & Shalev-Shwartz, 2017; Yi & Wu, 2019). A few works focus on further improving the sample selection quality by modelling the loss with markov process (Xia et al., 2021) or dynamically select samples with multiple metrics (Zhou et al., 2020). Instead of selecting samples based on the model prediction, some works try to utilize the feature representations for sample selection. Wu et al. (2020) and Wu et al. (2021) try to build a kNN graph and identify clean samples through connected sub-graphs, while Feng et al. (2022) and Ortego et al. (2021) propose to utilize a simpler kNN in feature space to alleviate the effect of noisy labels. Some recent methods involving contrastive learning also identify clean sample pairs based on neighborhood relationships in the feature space (Li et al., 2022) or fit Gaussian distributions to model the clean distribution (Huang et al., 2023). However, these methods remain unstable and prone to self-confirmation bias, especially in strong noise scenarios, due to its intrinsic reliance on the in-training model based on noisy dataset.

**Utilization of auxiliary model**  To alleviate self-confirmation bias, the utilization of an auxiliary noise-free model is reasonable and straightforward. Related to us, some methods also try to use pre-trained noise-free models for learning with noisy labels. Zheltonozhskii et al. (2021); Cheng et al. (2021) propose to utilize self-supervised pre-training since it can learn good representations in the label-free case. Bahri et al. (2020) utilize the pre-logit space of the pretrained model along with the kNN classifier for sample selection. Zhu et al. (2022) follows the same idea and also involves CLIP, but they only utilize its vision encoder as a common pretrained encoder without utilizing the language encoder. We emphasize that language modality is critical as a supplementary modality.

**Exploration of whole dataset**  To fully explore the entire dataset, specifically the non-selected subset, earlier methods propose label correction techniques (Zhou et al., 2020; Song et al., 2019). More recent methods typically treat the selected subset as labeled and the non-selected subset as unlabeled, applying existing semi-supervised learning methods. For instance, techniques like MixMatch (Berthelot et al., 2019) employed by DivideMix and FixMatch used by Self-Filter (Wei et al., 2022) fall into this category. Some loss functions that do not involve labels, such as contrastive loss, have also been applied to these samples to indirectly incorporate them into the training process. In this paper, we present a semi-supervised learning method tailored for noisy datasets called *MixFix*. Differing from existing methods, we set different thresholds based on the consistency between their model-predicted labels and the given noisy labels when introducing samples into training.

## 3 METHOD

In section 3.1, we cast the learning with noisy labels problem in a formulation that covers mainstream sample selection methods. We also provide essential details about the CLIP model. In section 3.2, we elaborate our sample selection method, namely *CLIPSelector*. In section 3.3, we introduce our semi-supervised learning method, namely *MixFix*. In section 3.4, we provide further discussions on the topics of sample selection and the use of the CLIP model for this purpose.

### 3.1 PROBLEM FORMULATION

**Sample selection with noisy labels**  Given a dataset of training samples $(\boldsymbol{x}_i, y_i)_{i=1}^N$ *i.i.d* sampled from a noisy joint distribution $P(\boldsymbol{x}, y)$ with support as $\sup(P) = \{\boldsymbol{x} \in \mathbb{R}^{C \times H \times W}, y \in \{1, ..., K\}\}$ where $K$ denotes the number of semantic classes, the goal of our method is to learn a classifier $f$ that can accurately predict the true labels $y$ for new, unseen examples. Let us denote the clean joint distribution as $P^{true}(\boldsymbol{x}, y)$. Most sample selection methods aim to approximate and optimize the unbiased empirical risk of $f$ on the clean joint distribution $P^{true}(\boldsymbol{x}, y)$ with samples from noisy joint distribution $P(\boldsymbol{x}, y)$: $\hat{R}^{true}(f) = \frac{1}{N} \sum_{i=1}^N w_i L(\boldsymbol{x}_i, y_i; f)$, where $w_i$ are the sample weights. Particularly, with optimal weights ($w_i = P^{true}(y_i|\boldsymbol{x}_i)/P(y_i|\boldsymbol{x}_i)$) we can achieve risk-consistent learning[2]. However, since $P^{true}(y_i|\boldsymbol{x}_i)$ and $P(y_i|\boldsymbol{x}_i)$ are typically both unknown for $\boldsymbol{x}_i$, the objective

---

[2]Refer to Appendix F for details. We omit the variables for brevity, e.g, $P(y = y_i|\boldsymbol{x} = \boldsymbol{x}_i)$ as $P(y_i|\boldsymbol{x}_i)$.

of sample selection methods often revolves around estimating these two to subsequently estimate the optimal weights. In general, the noisy label $y_i$ can serve as a confident proxy of the noisy distribution $P(y_i|\boldsymbol{x}_i)$, making our focus on utilizing an additional auxiliary classifier $\tilde{P}(y_i|\boldsymbol{x}_i)$ to estimate $P^{true}(y_i|\boldsymbol{x}_i)$. Here, we propose a concise form sufficient to comprehensively represent most existing sample selection methods:

$$\tilde{w}_i = \mathbb{G}(\tilde{P}(y_i|\boldsymbol{x}_i), y_i) \in 0, 1, \tag{1}$$

where $\mathbb{G}$ denotes a specific sample selection mechanism, such as the 'small loss' strategy, to further refining the estimation. In addition, it is commonly accepted to restrict the weights as binary since for most classification datasets, $P^{true}(y_i|\boldsymbol{x}_i)$ tends to be highly centered around only one class. As a result, the optimal weight $w_i$ usually leans towards either $0$ or $1$ for most samples.

**CLIP** We briefly introduce the CLIP model (Radford et al., 2021), which is currently one of the most prevalent vision-language models. CLIP aims to learn from a dataset of image-text pairs, denoted as $(\boldsymbol{x}'_i, \boldsymbol{z}_i)_{i=1}^M$, which is *i.i.d.* sampled from a hidden joint distribution $Q(\boldsymbol{x}, \boldsymbol{z})$ with support as $\sup(Q) = \{\boldsymbol{x} \in \mathbb{R}^{C \times H \times W}, \boldsymbol{z} \in \mathbb{R}^d\}$. We have below as CLIP training loss:

$$L(\boldsymbol{x}'_i, \boldsymbol{z}_i; g, h) = \frac{1}{2}(-\log \frac{\exp(g(\boldsymbol{x}'_i)^T h(\boldsymbol{z}_i))}{\sum_{j=1}^M \exp(g(\boldsymbol{x}'_i)^T h(\boldsymbol{z}_j))} - \log \frac{\exp(g(\boldsymbol{x}'_i)^T h(\boldsymbol{z}_i))}{\sum_{j=1}^M \exp(g(\boldsymbol{x}'_j)^T h(\boldsymbol{z}_i))}). \tag{2}$$

Here, $g$ and $h$ denote the vision and language encoder, respectively. Intuitively, the CLIP model tries to maximize the correspondence between positive image-text pairs.

### 3.2 CLIPSELECTOR: SAMPLE SELECTION WITH VISION-LANGUAGE MODELS

In this section, we propose a new sample selection method based on CLIP, namely *CLIPSelector*. According to eq. (1), our method is divided into two main steps: *1. estimate $\tilde{P}(y_i|\boldsymbol{x}_i)$ with CLIP; 2. calculate weight $\tilde{w}_i$ with specific $\mathbb{G}$*. For the convenience of subsequent presentations, we make the notations consistent for CLIP's training dataset and the in-question noisy dataset. Specifically, we extend the in-question noisy dataset to be sampled *i.i.d* from $P(\boldsymbol{x}, y, \boldsymbol{z})$, where $\sup(P) = \{\boldsymbol{x} \in \mathbb{R}^{C \times H \times W}, y \in [0, 1, \dots, K], \boldsymbol{z} \in \mathbb{R}^d\}$; similarly, we extend the sampling distribution of CLIP's training dataset to $Q(\boldsymbol{x}, y, \boldsymbol{z})$, where $\sup(Q) = \{\boldsymbol{x} \in \mathbb{R}^{C \times H \times W}, y \in 0, 1, \dots, K, \dots, K^\infty, \boldsymbol{z} \in \mathbb{R}^d\}$.

#### 3.2.1 ESTIMATE $\tilde{P}(y_i|\boldsymbol{x}_i)$ WITH CLIP

We consider two options for estimation: directly utilizing CLIP's zero-shot classifier, or, ignoring CLIP's language modality and treating its vision encoder as a regular pre-trained model and training a new classifier atop it with in-question noisy dataset.

**Estimate $\tilde{P}(y_i|\boldsymbol{x}_i)$ with CLIP zero-shot classifier** Firstly, we assume the causal mechanism for $P$ and $Q$ as: $\boldsymbol{x} \to \boldsymbol{z} \to y$ where $\boldsymbol{z}$ denotes the description and $y$ denotes the semantic label thus we have $y \perp \boldsymbol{x} \mid \boldsymbol{z}$. Roughly speaking, we assume that the semantic label $y_i$ can be independently generated based on a decent image description $\boldsymbol{z}_i$ alone for each image $\boldsymbol{x}_i$. We thus have:

$$\tilde{P}_{zeroshot}(y_i|\boldsymbol{x}_i) = \int Q(y_i|\boldsymbol{z}_i)Q(\boldsymbol{z}_i|\boldsymbol{x}_i)dz \propto \int Q(y_i|\boldsymbol{z}_i)Q(\boldsymbol{z}_i, \boldsymbol{x}_i)dz. \tag{3}$$

Specifically, according to eq. (2), we show that $Q(\boldsymbol{z}_i, \boldsymbol{x}_i)$ can be estimated by computing the output similarity $(\exp(g(\boldsymbol{x}_i)^T h(\boldsymbol{z}_i)))$ of the CLIP model (Appendix F). However, $Q(y_i|\boldsymbol{z}_i)$ remains unknown and cannot be learned during the CLIP training process. Most current studies customarily design a single prompt as follows: 'A photo of class name of $y_i$.', implicitly assuming that $Q(y = y_i|\boldsymbol{z} = $'A photo of class name of $y_i$.'$) \approx 1$. We can then estimate the integral by sampling $\boldsymbol{z}_i$. It is plausible that with more high-quality samplings of $\boldsymbol{z}_i$ instead of only utilizing one single prompt the estimation would be better. In this work, we propose below template to generate multiple prompts using class-specific features:

```
'A photo of {class name of yi}, which is/has {class-specific
                    feature j of class yi}.'
```

For more details about how to generate prompts, please refer to Appendix B. Then we can simplify eq. (3) with above prompts as below:

$$\tilde{P}_{zeroshot}(y_i|\boldsymbol{x}_i) \varpropto \sum_{j=1}^{J} \tilde{Q}(\boldsymbol{z} = \text{`A photo of \{class name of } y_i\}\text{, which}} \atop \text{is/has \{feature } j \text{ of class } y_i\}\text{.', } \boldsymbol{x}_i). \tag{4}$$

**Estimate $\tilde{P}(y_i|\boldsymbol{x}_i)$ with CLIP vision encoder and noisy dataset** By treating the CLIP model as an ordinary large-scale pre-trained model, we can leverage its vision encoder $g$ solely along with the in-question noisy dataset $(\boldsymbol{x}_i, y_i)_{i=1}^{N}$ to train a new classifier $f'$ for estimation. With the common cross-entropy loss, it is straightforward that the normalized prediction logits serve as an estimate of $\tilde{P}(y|\boldsymbol{x})$:

$$\tilde{P}_{trained}(y_i|\boldsymbol{x}_i) = \text{softmax}(f'(g(\boldsymbol{x}_i)))_{y_i}. \tag{5}$$

By default, we train a *LogisticRegression* classifier as $f'$ with fixed extracted features and noisy dataset. Empirically, we also consider non-parametric *kNN* in ablations Section 4.1.

### 3.2.2 CALCULATE WEIGHT $w_i$

With $\tilde{P}(y_i|\boldsymbol{x}_i)$ estimated above, we can estimate weight $w_i$ for each sample with any applicable sample selection mechanism $\mathbb{G}$. In this work, we consider two simple and popular mechanisms, named $\mathbb{G}_{loss}$ and $\mathbb{G}_{consistency}$. For $\mathbb{G}_{loss}$ , we firstly model the per-sample cross-entropy losses ($\{-\log \tilde{P}(y = y_i|\boldsymbol{x}_i)\}_{i=1}^{N}$) with GMM and then select samples by thresholding its probability belonging to the smaller component. Due to the possible class imbalances and the various semantic diversity of different classes, slightly different than the common approach utilizing a single GMM[3], we model the losses of samples from each class by a separate GMM model.

$$\mathbb{G}_{loss} = \mathbb{1}(\mathbb{P}(-\log \tilde{P}(y = y_i|\boldsymbol{x}_i) \in \text{GMM}_{small}) \geq \theta_{loss}). \tag{6}$$

For $\mathbb{G}_{consistency}$, we calculate a consistency measure (defined as the ratio of the probability of noisy label class to the highest class probability) and select samples with high consistency:

$$\mathbb{G}_{consistency} = \mathbb{1}(\tilde{P}(y = y_i|\boldsymbol{x}_i)/\max_{k} \tilde{P}(y = k|\boldsymbol{x}_i) \geq \theta_{cons}). \tag{7}$$

### 3.3 MIXFIX: EFFICIENT SEMI-SUPERVISED TRAINING BY ABSORBING AND RELABELLING

To evaluate our method on widely-acknowledged benchmarks, in this section, we propose a simple semi-supervised learning method for noisy dataset — namely *MixFix*. Please note, the notations employed in this section are defined independently. Specifically, we denote the selected subset and non-selected subset as $(\mathcal{X}_c, \mathcal{Y}_c)$ and $(\mathcal{X}_n, \mathcal{Y}_n)$ respectively. Motivated by pseudo-labelling (Lee et al., 2013) and FixMatch (Sohn et al., 2020), we then inspect each sample's current prediction $\boldsymbol{p}_i$ in non-selected subset with:

$$(w_i, y_i) = \begin{cases} (0, y_i), & \text{if } \max_{l} \boldsymbol{p}_i(l) < \theta_r \text{ and } \max_{l} \boldsymbol{p}_i(l) < \theta_r' & \texttt{*Drop*} \\ (1, y_i), & \text{if } \max_{l} \boldsymbol{p}_i(l) > \theta_r \text{ and } y_i = \arg\max_{l} \boldsymbol{p}_i(l) & \texttt{*Absorb*} \\ (1, \arg\max_{l} p_i(l)), & \text{if } \max_{l} \boldsymbol{p}_i(l) > \theta_r' \text{ and } y_i \neq \arg\max_{l} \boldsymbol{p}_i(l) & \texttt{*Relabel*} \end{cases} \tag{8}$$

Intuitively, we 'absorb' missed clean samples ($y_i = \arg\max_{l} \boldsymbol{p}_i(l)$) and 'relabel' noisy samples ($y_i \neq \arg\max_{l} \boldsymbol{p}_i(l)$) with different thresholds in non-selected subset, and progressively append it to initial selected subset to form a dynamic larger training set. Differing from existing semi-supervised learning techniques, we typically set $\theta_r \leq \theta_r'$. This helps us make full use of noisy labels to differentiate the 'absorb' and 'relabel' process. Please refer to Section 4.1 for more analysis. To further counter the class imbalance in this new training set, the minority class is over-sampled. Then, we apply a common cross-entropy loss for training with this expanded and class-balanced training set, along with Mixup interpolation (Zhang et al., 2017). The detailed process is presented in Algorithm 1.

### 3.4 ADDITIONAL DISCUSSION

---

[3]Please refer to Appendix C for more comparisons on seperate GMM and single GMM.

**To be greedy or conservative?** For all sample selection methods, an inevitable challenge is how to balance the precision and recall of sample selection. In this paper, we adopt a conservative sample selection strategy by taking the intersection of different sample selection outcomes, prioritizing the precision of sample selection. Compared to more greedy sample selection strategies, we lean towards relying on the semi-supervised learning strategy - *MixFix* to gradually introduce more samples into training. This can avoid magnifying the influence of noisy samples due to excessively greedy sample selection, but it also has obvious weaknesses, that some 'hard' clean samples will inevitably be missed.

---

**Algorithm 1: MixFix**.

**Input :** Selected subset $(\mathcal{X}_c, \mathcal{Y}_c)$, non-selected
        subset $(\mathcal{X}_n, \mathcal{Y}_n)$, $\theta_r, \theta'_r$, max epochs $T$
**while** $i < T$ **do**
    Generate $(\mathcal{X}_r^i, \mathcal{Y}_r^i)$ with eq. (8) ;
    Generate $(\mathcal{X}_t^i, \mathcal{Y}_t^i)$ with $(\mathcal{X}_r^i, \mathcal{Y}_r^i)$ and
    $(\mathcal{X}_c, \mathcal{Y}_c)$ ;
    Minority over-sampling with $(\mathcal{X}_t^i, \mathcal{Y}_t^i)$ ;
    Model training with $(\mathcal{X}_t^i, \mathcal{Y}_t^i)$ and MIXUP.
**end**

---

**To fully explore CLIP?** The utilization of the CLIP model for learning with noisy labels remains an area that requires further investigation. To ensure a fair comparison with existing work, we adopt standard sample selection paradigm, refraining from training or fine-tuning the CLIP model (Zhou et al., 2022; Chen et al., 2022). In addition to exploiting CLIP for sample selection, incorporating established techniques for learning from noisy labels into prompt-based learning may also offer promising directions, please refer to Appendix E for preliminary results.

## 4 EXPERIMENTS

In this section, we conduct extensive experiments on two standard benchmarks with synthetic label noise, CIFAR-10 and CIFAR-100, and four real-world noisy datasets, Red Mini-ImageNet (Jiang et al., 2020), Clothing1M (Xiao et al., 2015), WebVision (Li et al., 2017), and ANIMAL-10N (Song et al., 2019). We mainly follow previous works (Li et al., 2020a; Garg et al., 2023; Feng et al., 2022) for model structures and training settings, please refer to Appendix A for more details.

### 4.1 ABLATIONS STUDY

**Analyzing sample selection *w.r.t* different classifiers and different mechanisms** In appendix G, we theoretically conclude that the performance of the zero-shot classifier is influenced by the quality of utilized prompts and the domain gap between CLIP training dataset and the in-question noisy dataset, while the performance of the easily-inducible classifier trained based on CLIP's vision encoder and the in-question noisy dataset is influenced by the noise of the in-question dataset. To validate this, we empirically test with two datasets with controllable noise ratios, that is, the CIFAR10/100 dataset with synthetic noise and the Red Mini-ImageNet dataset with real-world noise. In fig. 2, we show the sample selection performance and find that: i) As the noise ratio increases, regardless of the dataset, noise types, the backbone of the CLIP model or the empirical variant of the trained classifier (*LogisticRegression* VS *kNN*), the zero-shot classifier gradually outperforms the trained classifier. This further validates our theoretical findings; ii) Additionally, we notice that when comparing two different modes for obtaining the training classifier, the *LogisticRegression* classifier empirically exhibits superior performance to the *kNN* classifier. Therefore, we choose the *LogisticRegression* classifier as our default choice for trained classifier; iii) Furthermore, we find that different sample selection mechanisms ($\mathbb{G}_{consistency}$ VS $\mathbb{G}_{loss}$) show distinct advantages and disadvantages on different datasets. Given that noise information is typically unknown in real-world scenarios, as analyzed in section 3.4, we default to a conservative sample selection strategy, which involves simultaneously utilizing multiple sample selection strategies and selecting their intersection.

**Analyzing CLIP Zero-shot classification as a baseline** In this section, we consider utilizing CLIP's zero-shot classifier directly with the test set, following a procedure that we describe in Section 3.2. In table 1, we present the zero-shot classification results on six common benchmarks and compare them with current state-of-the-art methods (best results summarized from subsequent tables in the paper.) as well as our own method. It's worth noting that CLIP is utilized with the VIT-B/32 architecture here, while our method and the SOTA methods adopt simpler structures, such as PreResNet-18 for the CIFAR dataset. Therefore, this comparison is indeed 'over stringent'.

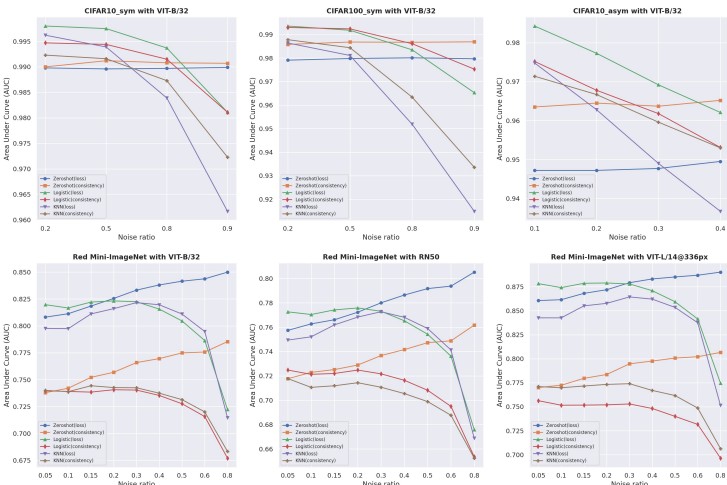

Figure 2: Comparisons of various sample selection methods *w.r.t* different dataset/noise type/noise ratio. Here, we show the ROC AUC score of binary identification of clean samples.

Still, we observe that, when compared to directly utilizing CLIP's zero-shot classifier, our method delivers significantly improvements on most datasets, with the exception of the Red Mini-ImageNet dataset. We attribute this to the fact that the Red Mini-ImageNet dataset introduces noisy samples collected through online search engines and replaces clean samples with specific proportions. As suggested in Theorem 1, this results in a small domain gap, which in turn benefits CLIP's performance. Nevertheless, our approach outperforms the SOTA on all datasets.

Table 1: Testing accuracy (%) with CLIP zero-shot classifier

| Model | CIFAR10 | CIFAR100 | Red Mini-ImageNet | WebVision | Clothing1M | ANIMAL-10N |
|---|---|---|---|---|---|---|
| CLIP zero-shot | 89.97 | 63.72 | 78.12 | 73.36 | 39.73 | 76.12 |
| SOTA | 92.68 | 67.7 | 49.55 | 80.9 | 74.84 | 84.6 |
| Ours | **95.15** | **71.17** | 54.21 | **81.56** | **74.87** | **88.14** |

**Sample selection with other vision-language models** Here, we compare CLIP with another vision-language model - ALIGN (Jia et al., 2021). Specifically, we compare their performance on sample selection based on the CIFAR10 dataset with instance-dependent noise (Chen et al., 2021a). In table 2, we can see ALIGN behaves similarly well as CLIP concerning precision with even higher recall. This demonstrates that our proposed idea of using vision-language models for sample selection is widely effective.

Table 2: Precision-Recall of sample selection results with CLIP and ALIGN.

| Noise ratio | 0.1 | | 0.2 | | 0.3 | | 0.4 | |
|---|---|---|---|---|---|---|---|---|
| | Precision | Recall | Precision | Recall | Precision | Recall | Precision | Recall |
| CLIP | 99.73 | 70.75 | 99.53 | 75.07 | 99.25 | 77.77 | 99.03 | 79.23 |
| ALIGN | 99.47 | 72.47 | 99.13 | 78.64 | 99.01 | 81.22 | 98.74 | 84.53 |

**Hyper-parameters *w.r.t MixFix*** In this section, we ablate on the only two hyperparameters of our semi-supervised training strategy *MixFix*: the 'absorb' threshold $\theta_r$ and the 'relabel' threshold $\theta_r'$. Similar to our dilemma when doing sample selection, here we also need to weigh the precision and recall when introducing additional training samples. In table 3 we demonstrate that under different noise ratios, a too high or too low threshold leads to performance degradation. In fig. 3, we further reveal the inherent mechanism, for example, after reducing the 'absorb' threshold $\theta_r'$, the proportion of training samples increases and the accuracy of training samples decreases.

Table 3: Ablations on *MixFix* with synthetic CIFAR100 noisy dataset. The *top-3* results are bolded.

| $\theta_r$ | $\theta_r'$ | Noise ratio | | | |
|---|---|---|---|---|---|
| | | 20% | 50% | 80% | 90% |
| | 0.7 | 76.46 | 74.69 | **69.50** | 62.91 |
| 0.7 | 0.8 | **76.63** | **75.23** | **69.72** | **63.11** |
| | 0.9 | **77.06** | **75.17** | 67.76 | 59.17 |
| | 0.7 | 75.49 | 74.30 | 67.95 | **63.29** |
| 0.8 | 0.8 | 76.36 | **74.90** | 68.86 | **63.42** |
| | 0.9 | **76.66** | 74.50 | 67.37 | 58.09 |
| | 0.7 | 74.53 | 73.49 | 68.74 | 62.22 |
| 0.9 | 0.8 | 75.98 | 74.25 | **68.94** | 62.81 |
| | 0.9 | 75.78 | 74.23 | 67.17 | 59.38 |

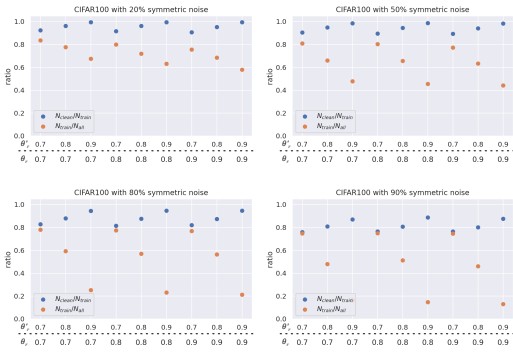

Figure 3: $N_{train}$ denotes number of training samples, $N_{clean}$ denotes number of clean training samples and $N_{all}$ denotes number of clean training samples.

## 4.2 RESULTS ON SYNTHETIC NOISY DATASET

In this section, we firstly evaluate our method on the CIFAR datasets with synthetic symmetric/asymmetric noise. In table 4, We can see that our method gets competitive and better performance in all experiment settings, especially when the noise ratio is high (63.11% testing accuracy with 90% symmetric noise on CIFAR100 dataset). Also, we would like to emphasize that we keep hyper-parameters fixed for all experiments here as we believe the method robustness in a noise agnostic scenario is critical.

Table 4: Testing accuracy (%) on on CIFAR-10 and CIFAR-100 with synthetic noise.

| Dataset | CIFAR10 | | | | | CIFAR100 | | | |
|---|---|---|---|---|---|---|---|---|---|
| Noise type | Symmetric | | | | Assymetric | Symmetric | | | |
| Noise ratio | 20% | 50% | 80% | 90% | 40% | 20% | 50% | 80% | 90% |
| Cross-Entropy | 86.8 | 79.4 | 62.9 | 42.7 | 85.0 | 62.0 | 46.7 | 19.9 | 10.1 |
| Co-teaching+ (Yu et al., 2019) | 89.5 | 85.7 | 67.4 | 47.9 | - | 65.6 | 51.8 | 27.9 | 13.7 |
| F-correction (Patrini et al., 2017) | 86.8 | 79.8 | 63.3 | 42.9 | 87.2 | 61.5 | 46.6 | 19.9 | 10.2 |
| PENCIL (Yi & Wu, 2019) | 92.4 | 89.1 | 77.5 | 58.9 | 88.5 | 69.4 | 57.5 | 31.1 | 15.3 |
| LossModelling (Arazo et al., 2019) | 94.0 | 92.0 | 86.8 | 69.1 | 87.4 | 73.9 | 66.1 | 48.2 | 24.3 |
| DivideMix (Li et al., 2020a) | **96.1** | 94.6 | 93.2 | 76.0 | 93.4 | 77.3 | 74.6 | 60.2 | 31.5 |
| ELR+ (Liu et al., 2020) | 95.8 | 94.8 | 93.3 | 78.7 | 93.0 | 77.6 | 73.6 | 60.8 | 33.4 |
| MOIT (Ortego et al., 2021) | 93.1 | 90.0 | 79.0 | 69.6 | 92.0 | 73.0 | 64.6 | 46.5 | 36.0 |
| SelCL+ (Li et al., 2022) | 95.5 | 93.9 | 89.2 | 81.9 | 93.4 | 76.5 | 72.4 | 59.6 | 48.8 |
| TCL (Huang et al., 2023) | 95.0 | 93.9 | 92.5 | 89.4 | 92.6 | 78.0 | 73.3 | 65.0 | 54.5 |
| Ours | 95.92±0.15 | 95.67±0.28 | 95.04±0.37 | 94.23±0.54 | 94.89±0.16 | 78.20±0.45 | 75.23±0.29 | 69.72±0.61 | 63.11±0.89 |

To further validate the performance of our method in handling the 'hard noise', we also conduct experiments on instance-dependent noise in table 5. Different from symmetric or asymmetric noise, instance-dependent noise assumes that semantic-similar samples are more prone to get mislabelled, aligning better with our earlier definition of 'hard noise'. Besides, here we here exclude *MixFix* and solely employ the selected samples for training with cross-entropy loss. This exclusion serves to provide an additional ablation analysis of the sample selection performance of *CLIPSelector*.

## 4.3 RESULTS ON REAL-WORLD NOISY DATASETS

Finally, in table 6, table 7, and table 8 we show results on the ANIMAL-10N, Red Mini-ImageNet and WebVision datasets, respectively. In summary, our proposed method demonstrates substantial improvements compared to the current state-of-the-art approaches on both large-scale web-crawled datasets and small-scale human-annotated noisy datasets. We note, that the proposed sample selection method can be used in combination with other schemes. In table 9 we show results on the Clothing1M dataset both with our default setting (*CLIPSelector + MixFix*) and with it incorporated to two additional schemes. First incorporating our method with co-training, and second replacing *MixFix* with DivideMix (Li et al., 2020a). We observe that we obtain results that are superior to the current state-of-the-art. However, we would like to note that the majority of existing methods have small

Table 5: Testing accuracy (%) on CIFAR10 with instance-dependent noise.

| Method | Noise ratio | | | |
|---|---|---|---|---|
| | 10% | 20% | 30% | 40% |
| Cross-Entropy | 91.25 | 86.34 | 80.87 | 75.68 |
| F-correction (Patrini et al., 2017) | 91.06 | 86.35 | 78.87 | 71.12 |
| Co-teaching (Han et al., 2018) | 91.22 | 87.28 | 84.33 | 78.72 |
| GCE (Zhang & Sabuncu, 2018) | 90.97 | 86.44 | 81.54 | 76.71 |
| DAC (Thulasidasan et al., 2019) | 90.94 | 86.16 | 80.88 | 74.80 |
| DMI (Xu et al., 2019) | 91.26 | 86.57 | 81.98 | 77.81 |
| SEAL (Chen et al., 2021a) | 91.32 | 87.79 | 85.30 | 82.98 |
| Cross-Entropy* | 90.76 | 86.08 | 80.64 | 75.27 |
| CLIPSelector + Cross-Entropy | **92.33±0.37** | **91.06±0.37** | **89.71±0.37** | **88.26±0.37** |

Table 6: Testing accuracy (%) on ANIMAL-10N.

| Method | Accuracy |
|---|---|
| Cross-Entropy | 79.4 |
| SELFIE (Song et al., 2019) | 81.8 |
| PLC (Zhang et al., 2021b) | 83.4 |
| NCT (Chen et al., 2021b) | 84.1 |
| InstanceGM (Garg et al., 2023) | 84.6 |
| Ours | **88.14±0.46** |

differences on the Clothing1M dataset despite the fact that they have large performance differences on other datasets. This suggests that additional training techniques may have a greater impact than sample selection methods on this specific dataset, possibly due to the fact that the Clothing1M dataset is more fine-grained than other datasets. For such fine-grained noisy datasets, sample selection may not be the optimal strategy, as suggested in Section 3.1, where the basis of sample selection methods relies on highly concentrated conditional probabilities for the samples.

Table 7: Testing accuracy (%) on on Red Mini-ImageNet.

| Method | Noise ratio | | | |
|---|---|---|---|---|
| | 20% | 40% | 60% | 80% |
| Cross-Entropy | 47.36 | 42.70 | 37.30 | 29.76 |
| Mixup (Zhang et al., 2017) | 49.10 | 46.40 | 40.58 | 33.58 |
| DivideMix (Li et al., 2020a) | 50.96 | 46.72 | 43.14 | 34.50 |
| MentorMix (Jiang et al., 2020) | 51.02 | 47.14 | 43.80 | 33.46 |
| FaMUS (Xu et al., 2021) | 51.42 | 48.06 | 45.10 | 35.50 |
| InstanceGM (Garg et al., 2023) | 58.38 | 52.24 | 47.96 | 39.62 |
| Ours | **61.44±0.45** | **58.42±0.66** | **53.18±0.47** | **43.82±0.87** |

Table 8: Testing accuracy (%) on on WebVision.

| Methods | WebVision | | ILSVRC2012 | |
|---|---|---|---|---|
| | Top1 | Top5 | Top1 | Top5 |
| Co-teaching (Han et al., 2018) | 63.5 | 85.20 | 61.48 | 84.70 |
| DivideMix (Li et al., 2020a) | 77.32 | 91.64 | 75.20 | 90.84 |
| ELR+ (Liu et al., 2020) | 77.78 | 91.68 | 70.29 | 89.76 |
| NGC (Wu et al., 2021) | 79.16 | 91.84 | 74.44 | 91.04 |
| FaMUS (Xu et al., 2021) | 79.4 | 92.8 | 77.0 | 92.8 |
| RRL (Li et al., 2020b) | 76.3 | 91.5 | 73.3 | 91.2 |
| SelCL+ (Li et al., 2022) | 79.9 | 92.6 | 76.8 | **93.0** |
| SSR+ (Feng et al., 2022) | 80.9 | 92.8 | 75.8 | 91.8 |
| TCL (Huang et al., 2023) | 79.1 | 92.3 | 75.4 | 92.4 |
| Ours | **81.56±0.29** | **93.26±0.65** | **77.80±0.25** | 92.08±0.44 |

Table 9: Testing accuracy (%) on Clothing1M.

| Method | Accuracy |
|---|---|
| Cross-Entropy | 69.21 |
| F-correction (Patrini et al., 2017) | 69.84 |
| RRL (Li et al., 2020b) | 74.30 |
| C2D (Zheltonozhskii et al., 2021) | **74.84** |
| DivideMix (Li et al., 2020a) | 74.76 |
| ELR+ (Liu et al., 2020) | 74.81 |
| SSR+ (Feng et al., 2022) | 74.83 |
| TCL (Huang et al., 2023) | 74.80 |
| Ours | 73.41±0.65 |
| Ours (Co-training) | 74.01±0.47 |
| CLIPSelector + DivideMix | **74.87±0.44** |

## 5 CONCLUSION

To mitigate the issues of 'self-confirmation bias' and compensate for visual-only modality in current mainstream sample selection methods, in this paper we propose a method utilizing the large-scale vision-language model CLIP for sample selection, called *CLIPSelector*. We substantiate its effectiveness through both theoretically and empirically. Furthermore, we introduce a straightforward semi-supervised learning method tailored for noisy datasets, called *MixFix*, without the need for intricate off-the-shelf techniques. We emphasize that the exploration of utilizing vision-language models for noisy datasets, such as the potential of existing prompt learning techniques, remains an open direction. Additionally, the possibility of a large domain gap between the CLIP model and the target dataset can influence results, indicating a need for more refined vision-language models. Lastly, our experiments suggest that sample selection methods may not be optimal for fine-grained noisy datasets, which presents itself also as one of our future research directions.

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

## A    EXPERIMENT DETAILS

In this section, we present the dataset details and implementation details.

### A.1    DATASET DETAILS

**CIFAR10** and **CIFAR100** datasets comprise 50,000 images. Following established conventions, we assess our method's performance with two types of artificial noise: "symmetric noise," wherein labels are randomly flipped across all samples using a uniform distribution, and "asymmetric noise," wherein labels of visually similar categories, such as Horse $\leftrightarrow$ Deer and Dog $\leftrightarrow$ Cat, are randomly interchanged. Moreover, we conduct experiments with various noise levels: 20%, 50%, 80% and 90% symmetric noise, as well as 40% asymmetric noise, adhering to the settings in DivideMix ((Li et al., 2020a)). For instance-dependent noise, we utilize the label noise file provided by Chen et al. (2021a) (https://github.com/chenpf1025/IDN/tree/master/data/CIFAR10/label_noisy).

**Red Mini-ImageNet** dataset (Jiang et al., 2020) is a real-world dataset containing a total of 100 categories. It is an extension of the Mini-Imagenet dataset, where noise is introduced at varying ratios. Specifically, noisy images and their respective labels are obtained by crawling the internet, and these noisy images replace the original images in the Mini-ImageNet dataset, with different noise ratios. To ensure a fair comparison with previous studies (Garg et al., 2023; Xu et al., 2021), the images are resized from their original size of 84×84 pixels to 32×32 pixels. Moreover, in accordance with the existing literature (Garg et al., 2023; Xu et al., 2021), we utilize noise ratios of 20%, 40%, 60%, and 80%.

**WebVision** (Li et al., 2017) is an extensive dataset comprising 1,000 classes of images obtained through web crawling. In line with previous studies (Jiang et al., 2018; Li et al., 2020a; Ortego et al.,

2021), we evaluate our methods using the top 50 classes from the Google Subset of WebVision. The estimated noise ratio for this subset is approximately 20%.

**ANIMAL-10N** (Song et al., 2019) is a recently introduced real-world noisy dataset comprises 10 classes of animals. The dataset has undergone manual labeling, with an estimated label noise ratio of around 8%. Similar to the CIFAR datasets, ANIMAL-10N consists of 50,000 training images and 10,000 test images.

**Clothing1M** (Xiao et al., 2015) is a large-scale dataset containing 14 classes of clothing images, obtained by crawling online shopping websites. It consists of a substantial collection of 1 million noisy images. The estimated noise ratio for this dataset is approximately 38.5%.

## A.2 IMPLEMENTATION DETAILS

We use CLIP model with VIT-B/32 backbone in all experiments except for specific ablations. In all experiments, our default approach is *CLIPSelector + MixFix* (**Ours**).

For **CIFAR10** and **CIFAR100**, we use a PresActResNet-18 (He et al., 2016) as the backbone in all experiments following previous works. For CIFAR10, we set $\theta_{loss} = 0.5, \theta_{cons} = 0.8$ for *CLIPSelector* and $\theta_r = 0.8, \theta_r' = 0.9$ for *MixFix*; For CIFAR10, we set $\theta_{loss} = 0.5, \theta_{cons} = 0.8$ for *CLIPSelector* and $\theta_r = 0.7, \theta_r' = 0.8$ for *MixFix*. We train both networks with the a SGD optimizer for 300 epochs with a momentum of 0.9 and a weight decay of 5e-4. The initial learning rate is 0.02 and is controlled by a cosine annealing scheduler. The batchsize is fixed as 128.

For **Red Mini-ImageNet**, we also use a PresActResNet-18 (He et al., 2016) as the backbone following previous works (Garg et al., 2023; Xu et al., 2021). For *CLIPSelector*, we set $\theta_{loss} = 0.5, \theta_{cons} = 0.8$. For *MixFix*, we set $\theta_r = 0.8, \theta_r' = 0.95$. We train the network with a SGD optimizer for 300 epochs with a momentum of 0.9 and a weight decay of 5e-4. The initial learning rate is 0.02 and reduced by a factor of 10 after 200 and 250 epochs. The batchsize is fixed as 64.

For **WebVision**, we use a InceptionResNetv2 as the backbone following (Li et al., 2020a). For *CLIPSelector*, we set $\theta_{loss} = 0.5, \theta_{cons} = 1$. For *MixFix*, we set $\theta_r = 0.7, \theta_r' = 1.0$. We train the network with a SGD optimizer for 150 epochs with a momentum of 0.9 and a weight decay of 1e-4. The initial learning rate is 0.01 and reduced by a factor of 10 after 80 and 120 epochs. The batchsize is fixed as 32.

For **Clothing1M**, we use a ResNet50 as the backbone following (Li et al., 2020a) with ImageNet pretrained weights. For *CLIPSelector*, we set $\theta_{loss} = 0, \theta_{cons} = 0.5$. For *MixFix*, we set $\theta_r = 0.7, \theta_r' = 1.0$. We train the network with a SGD optimizer for 150 epochs with a momentum of 0.9 and weight decay of 1e-3. The initial learning rate is 0.002 and reduced by a factor of 10 after 50 and 100 epochs. The batchsize is fixed as 32.

For **ANIMAL-10N**, we use a VGG-19 (Simonyan & Zisserman, 2014) as the backbone with batch-normalization following (Song et al., 2019). For *CLIPSelector*, we set $\theta_{loss} = 0.5, \theta_{cons} = 0.8$. For *MixFix*, we set $\theta_r = 0.7, \theta_r' = 0.95$. We train the network with SGD optimizer for 300 epochs with a momentum of 0.9 and weight decay of 5e-4. The initial learning rate is 0.02 and reduced by a factor of 10 after 150 and 250 epochs. The batchsize is fixed as 128.

## A.3 MIXUP

Mixup (Zhang et al., 2017) is widely used in current works dealing with label noise because it is very lightweight and easy to implement. Assuming two random samples $x_1, y_1$ and $x_2, y_2$, a mixed new sample $x_m, y_m$ will be generated as:

$$\lambda \sim Beta(\alpha, \alpha)$$
$$\lambda' = \max(\lambda, 1 - \lambda)$$
$$x_m = \lambda' x_1 + (1 - \lambda') x_2$$
$$y_m = \lambda' y_1 + (1 - \lambda') y_2.$$

We then train with the new virtual mixed sample $(x_m, y_m)$. Instead of direct training with samples from the clean subset, we expect that virtual samples generated by Mixup are further away from the dataset samples thus can alleviate the noise memorization effect (Zhang et al., 2017). in

our experiments, following DivideMix (Li et al., 2020a), we set $\alpha$ as $4$ for beta mixture for the CIFAR10/CIFAR100 datasets, and as $0.5$ for all other real-world noisy dataset.

## B  PROMPTS GENERATION AND FURTHER ANALYSIS

**How multiple prompts with class-specific features are generated?**  Regarding the generation multiple prompts based on class-specific features, motivated by recent work (Menon & Vondrick, 2022), we first generate multiple features for each class by asking ChatGPT about each category's characteristics. We use below question for ChatGPT 3.5:

```
For CLIP model, the prompts matter a lot.  can you give me some
discriminative features of some classes?  Please list it in nested
 python as each class has multiple descriptions.  Please ensure it
  is formatted as 'which has ...'  or 'which is ...'  or 'which
  ...'.  For example, ['Cat', 'Lynx', 'Wolf', 'Coyote', 'jaguar',
  'Cheetah', 'Chimpanzee', 'Orangutan', 'Hamster', 'Guinea pig'].
```

We then generate multiple prompts with template in Section 3.2. We will include our generated class-specific prompts along with the code upon acceptance.

**Comparison of class-specific prompts with other prompt style**  To experimentally validate the superiority of our prompt style based on class-specific features, we conduct a comparative analysis of its zero-shot classification performance against alternative prompt styles. Specifically, we consider three empirical variants including ours:

1. Single prompt: 'A photo of {class name of $y_i$}.';
2. Multiple prompts with different templates: 'A good photo of {class name of $y_i$}.'/'An old picture of {class name of $y_i$}.' .etc;
3. Multiple prompts with class-specific features: 'A photo of {class name of $y_i$}, which is/has {class-specific feature $j$ of class $y_i$}.' with features such as the color, shape, etc.

In table 10, we present zero-shot classification results on six noisy datasets using the three prompt styles mentioned above and different backbones for CLIP model (VIT-B/32 and VIT-L/14@336px). We observe that, in most cases, the effectiveness of our prompting style is at its best, especially when employing a larger-scale CLIP backbone (VIT-L/14@336px). This aligns with our theoretical analysis.

Table 10: Zero-shot classification with different prompt styles.

| Model | Prompt technique | CIFAR10 | CIFAR100 | Red Mini-ImageNet | WebVision | Clothing1M | ANIMAL-10N |
|---|---|---|---|---|---|---|---|
| | 1 | 88.29 | 61.62 | 74.40 | 72.40 | 39.80 | 75.08 |
| CLIP (ViT-B/32) | 2 | 89.73 | 63.65 | 75.14 | 68.12 | 39.68 | 75.70 |
| | 3 | 87.97 | 63.72 | 78.12 | 73.36 | 37.73 | 74.62 |
| | 1 | 94.78 | 74.36 | 80.20 | 45.13 | 85.18 | 85.12 |
| CLIP (ViT-L/14@336px) | 2 | 95.17 | 74.96 | 79.88 | 47.26 | **85.78** | 87.00 |
| | 3 | **95.19** | **76.78** | **81.96** | **48.15** | 85.36 | **87.98** |

## C  PER-CLASS SEPERATE GMM VS WHOLE SINGLE GMM

In this section, we compare the differences between using seperate GMM for each class and a single GMM for all classes in sample selection. We conduct experiments on the CIFAR10 dataset with instance-dependent noise. As shown in table 11, we observe that the seperate GMM yields a higher recall while maintaining competitive precision in sample selection. In table 12, we find and validate that the seperate GMM allows us to obtain a more balanced subset, thereby mitigating class imbalance issues and partially explaining why we achieve a better recall above.

Table 11: Precision and recall of sample selection on CIFAR10 dataset with instance-dependent noise with Separate and Single GMM.

| Noise ratio | 0.1 | | 0.2 | | 0.3 | | 0.4 | |
|---|---|---|---|---|---|---|---|---|
| | Precision | Recall | Precision | Recall | Precision | Recall | Precision | Recall |
| Separate GMM | 99.73 | 70.75 | 99.53 | 75.09 | 99.25 | 77.77 | 99.04 | 79.26 |
| Single GMM | 99.77 | 68.90 | 99.61 | 71.88 | 99.43 | 73.68 | 99.29 | 72.67 |

Table 12: Max-Min number of selected samples from each class.

| Noise ratio | 0.1 | | 0.2 | | 0.3 | | 0.4 | |
|---|---|---|---|---|---|---|---|---|
| | Max | Min | Max | Min | Max | Min | Max | Min |
| Separate GMM | 4061 | 2228 | 3938 | 2148 | 3656 | 1851 | 3312 | 1440 |
| Single GMM | 4188 | 1720 | 4038 | 1757 | 3682 | 1455 | 3403 | 947 |

## D  ADDITIONAL RESULTS WITH MODEL CO-TRAINING

Previous experiments demonstrated that our approach is simple and achieves competitive results, and we would also like to emphasize that our approach is not mutually exclusive but can be used in conjunction with existing techniques. For example, we can seamlessly use CLIPSelector with any existing method (including the mentioned UNICON and PES) by adding an additional warm-up stage using only selected samples in CLIPSelector. Here, as an illustration, we co-train the models (a proven effective and simple technique by exchanging selected samples between the two models) on the CIFAR100 and Animal-10N datasets after combining it with our method Results are presented (table 13), further demonstrating that our approach has great potential alongside existing techniques and is comparable to existing works.

Table 13: Incorporating model co-training with our method.

| Dataset | CIFAR100 0.5sym | CIFAR100 0.9sym | Animal-10N |
|---|---|---|---|
| Ours | 75.23 | 63.11 | 88.14 |
| Ours + Co-training | 77.51 | 66.72 | 88.79 |
| UNICON (Karim et al., 2022) | 77.6 | 44.8 | / |
| PES (Bai et al., 2021) | 74.3 | / | / |

## E  UTILIZING CLIP AS A PRETRAINED VISUAL ENCODER

In addition to applying the CLIP model for sample selection, as a pre-trained foundation model, combining CLIP with existing noisy label learning techniques is an interesting research direction that we intend to explore in the future. Here, we present preliminary experiments based on SSR (Feng et al., 2022) by replacing the original encoder with the CLIP pre-trained encoder. As shown in table 14, the pre-trained CLIP encoder provides effective improvement, further demonstrating the potential of the CLIP model for LNL. Also, please note that fine-tuning CLIP is not trivial, as described in the following GitHub issues. Through better hyperparameter settings, we believe there is still room for further progress.

## F  FULL DERIVATION IN SECTION 3.1 AND SECTION 3.2

In this section, we first provide the full derivation of the weighted empirical risk and the solution of optimal weight. We then briefly explain the relation of output similarity in CLIP model and the joint probability.

**Weighted empirical risk minimization in sample selection**   For better clarification, we here repeat the problem formulation in Section 3.1. Given a dataset of training samples $(x_i, y_i)_{i=1}^{N}$ *i.i.d* sampled

Table 14: Replacing encoder in SSR (Feng et al., 2022) with CLIP vision encoder.

| Method | CIFAR10 50% symmetric noise | CIFAR10 90% symmetric noise |
|---|---|---|
| SSR | 95.45 | 93.45 |
| CLIP+SSR | 96.25 | 95.98 |

from a hidden joint distribution $P(\boldsymbol{x}, y)$ with $supp(P) = \{\boldsymbol{x} \in R^{C \times H \times W}, y \in \{1, ..., K\}\}$ and $K$ denotes the number of semantic classes, the goal of supervised learning is to learn a model $f$ that can accurately predict the true labels $y$ for new, unseen examples. Mathematically, we often optimize the empirical risk with samples *i.i.d* sampled from noisy distribution $P(\boldsymbol{x}, y)$:

$$\hat{R}^P(f) = \frac{1}{N} \sum_{i=1}^{N} L(\boldsymbol{x}_i, y_i; f)$$

Here $L$ can be any applicable classification-calibrated surrogate loss to *0-1 loss*, normally we use *Cross-Entropy loss*:

$$L(\boldsymbol{x}_i, y_i; f) = -\log \frac{\exp(f(\boldsymbol{x}_i)_{y_i})}{\sum_{j=1}^{K} \exp(f(\boldsymbol{x}_i)_j)}.$$

Owing to the *ERM principle*, we can uniformly minimize *w.r.t* the expected risk by minimizing above empirical risk:

$$R^P(f) = E_{P(\boldsymbol{x},y)} L(\boldsymbol{x}, y; f)$$

However, in this work we focus on learning with noisy labels, that is to say, there exist discrepancy between the noisy training distribution $P(\boldsymbol{x}, y)$ and clean unknown distribution $P^{true}(\boldsymbol{x}, y)$. In this condition, for the same specific model $f$, we have the expected risk on real distribution as:

$$R^{true}(f) \triangleq R^{P^{true}}(f) = E_{P^{true}(\boldsymbol{x},y)} L(\boldsymbol{x}, y; f)$$

To bridge the distribution discrepancy, we can easily find that:

$$R^{true}(f) = E_{P^{true}(\boldsymbol{x},y)} L(\boldsymbol{x}, y; f) = E_{P(\boldsymbol{x},y)} \frac{P^{true}(\boldsymbol{x}, y)}{P(\boldsymbol{x}, y)} L(\boldsymbol{x}, y; f).$$

Further, we assume $\frac{P^{true}(\boldsymbol{x},y)}{P(\boldsymbol{x},y)} = \frac{P^{true}(y|\boldsymbol{x}) P^{true}(\boldsymbol{x})}{P(y|\boldsymbol{x}) P(\boldsymbol{x})} = \frac{P^{true}(y|\boldsymbol{x})}{P(y|\boldsymbol{x})}$ as label noise normally does not affect the sample itself ($P(\boldsymbol{x}) = P^{true}(\boldsymbol{x})$). We then get the corresponding weighted empirical risk with noisy labels,

$$\hat{R}^{true}(f) = \frac{1}{N} \sum_{i=1}^{N} \frac{P^{true}(y_i|\boldsymbol{x}_i)}{P(y_i|\boldsymbol{x}_i)} L(\boldsymbol{x}_i, y_i; f)$$

with which we can ensure a risk-consistent classifier *w.r.t* clean distribution learned with even noisy labels.

*More than sample selection?* Other than sample selection, another applicable direction is the so-called *risk-consistent* methods, for example, to estimate the noise transition matrix by assuming $P^{true}(y^{true}|\boldsymbol{x}) = T(y^{true}|y) P(y|\boldsymbol{x})$. A common assumption here is to assume the noise transition is instance-independent and label-dependent only thus to alleviate it from $T(y^{true}|y, \boldsymbol{x})$ to $T(y^{true}|y)$. Please refer to related paper (Xia et al., 2019; 2022) for more details. Though theoretically consistent, these methods often achieves relative sub-optimal performance, since noise modes in real-world datasets are extremely complex, and current noise models cannot accurately simulate them.

**Relation of joint probability $Q(\boldsymbol{x}_i, \boldsymbol{z}_i)$ and the CLIP similarity $\exp(g(\boldsymbol{x}_i)^T h(\boldsymbol{z}_i))$** The zero-shot classification paradigm in eq. (4) is widely applied, however without clear theoretical explanation. In this work, we bridge the CLIP model with zero-shot classification by eq. (3). We here explain

the probabilistic relation of the learned similarity value $(\exp(g(\boldsymbol{x}_i)^T h(\boldsymbol{z}_i)))$ and the joint probability $Q(\boldsymbol{x}_i, \boldsymbol{z}_i)$. Specifically, we can easily write the empirical risk with CLIP loss function in eq. (2) as:

$$\hat{R}^Q(g,h) = \frac{1}{2}\sum_{i=1}^{M}(-\log\frac{\exp(g(\boldsymbol{x}_i)^T h(\boldsymbol{z}_i))}{\sum_{j=1}^{M}\exp(g(\boldsymbol{x}_j)^T h(\boldsymbol{z}_i))} - \log\frac{\exp(g(\boldsymbol{x}_i)^T h(\boldsymbol{z}_i))}{\sum_{j=1}^{M}\exp(g(\boldsymbol{x}_i)^T h(\boldsymbol{z}_j))})$$

$$= -\frac{1}{2}\log\prod_{i=1}^{M}\frac{\exp(g(\boldsymbol{x}_i)^T h(\boldsymbol{z}_i))}{\sum_{j=1}^{M}\exp(g(\boldsymbol{x}_j)^T h(\boldsymbol{z}_i))}\frac{\exp(g(\boldsymbol{x}_i)^T h(\boldsymbol{z}_i))}{\sum_{j=1}^{M}\exp(g(\boldsymbol{x}_i)^T h(\boldsymbol{z}_j))}$$

For the specific *i.i.d* sampled dataset, based on *MLE principle* we have the negative log-likelihood as:

$$\mathcal{L}(g,h;(\boldsymbol{x}_i,\boldsymbol{z}_i)_{i=1}^{M}) = -\log\prod_{i=1}^{M}Q_{g,h}(\boldsymbol{x}_i|\boldsymbol{z}_i,\boldsymbol{x}\in\{\boldsymbol{x}_j\}_{j=1}^{M};g,h)Q_{g,h}(\boldsymbol{z}_i|\boldsymbol{x}_i,\boldsymbol{z}\in\{\boldsymbol{z}_j\}_{j=1}^{M};g,h)$$

$$= -\log\prod_{i=1}^{M}\frac{Q_{g,h}(\boldsymbol{x}_i,\boldsymbol{z}_i)}{\sum_{j=1}^{M}Q_{g,h}(\boldsymbol{x}_j,\boldsymbol{z}_i)}\frac{Q_{g,h}(\boldsymbol{x}_i,\boldsymbol{z}_i)}{\sum_{j=1}^{M}Q_{g,h}(\boldsymbol{x}_i,\boldsymbol{z}_j)}$$

Comparing $\hat{R}^Q(g,h)$ with $\mathcal{L}(g,h;(\boldsymbol{x}_i,\boldsymbol{z}_i)_{i=1}^{M})$, we have: $\exp(g(\boldsymbol{x}_i)^T h(\boldsymbol{z}_j)) \propto Q_{g,h}(\boldsymbol{x}_i,\boldsymbol{z}_j)$, where latter serves as an estimation of $Q(\boldsymbol{x}_i,\boldsymbol{z}_j)$ after training.

# G  THEORETICAL COMPARISON OF TWO OPTIONS FOR SAMPLE SELECTION WITH CLIP

An immediate question is: how does the zero-shot classifier (eq. (4)) compare to the trained classifier (eq. (5)) in estimating $\tilde{P}(y|x)$. If the latter demonstrates comparable or even superior performance to the former, there may be little incentive to employ the CLIP model for sample selection. Rather, pursuing further enhancements to existing large-scale visual-only pre-trained models may yield greater potential. To this end, we conduct a theoretical analysis and compare the distances between the estimated $\tilde{P}(y_i|\boldsymbol{x}_i)$ and true $P^{true}(y_i|\boldsymbol{x}_i)$ of the two options. Specifically, we have below theorems:

**Theorem 1** (ESTIMATION WITH ZERO-SHOT CLASSIFIER). *Let $\mathcal{G}, \mathcal{H}$ be the hypothesis space of vision encoder $g$ and language encoder $h$. Let us denote the rademacher complexity as $\Re(\mathcal{G}\circ\mathcal{H})$ of the combined CLIP model. Supposing the range of $L$ from eq. (2) as $[0, l_\infty^{clip}]$ for all $(\boldsymbol{x},\boldsymbol{z})$ in $\sup(Q)$ with $g,h\in\mathcal{G},\mathcal{H}$. Then, for any $\delta > 0$, with probability at least $1-\delta$ we have the following holds:*

$$d(\tilde{P}_{zeroshot}(y_i|\boldsymbol{x}_i), P^{true}(y_i|\boldsymbol{x}_i)) \le \varepsilon_{domain} + \Delta(\ \lambda_1\Re(\mathcal{G}\circ\mathcal{H}) + \lambda_2 l_\infty^{clip}\sqrt{\frac{\log 1/\delta}{M}} + \lambda_3\varepsilon_n)$$

*with $\lambda_1, \lambda_2, \lambda_3 > 0$. Here, $\varepsilon_{domain}$ denotes the bias term induced by the domain gap between $Q$ and $P^{true}$, and $\Delta \ge 1$ denotes the bias coefficient induced in designing prompts and sampling in eq. (3).*

**Theorem 2** (ESTIMATION WITH TRAINED CLASSIFIER ). *Let $\mathcal{F}$ be the hypothesis space of trained classifier $f'$. Let us denote the rademacher complexity as $\Re(\mathcal{F})$ of the trained classifier. Supposing the range of $L$ for training $f'$ as $[0, l_\infty^{noisy}]$ for all $(\boldsymbol{x},y)$ in $\sup(P)$ with $f'\in\mathcal{F}$. Then, for any $\delta > 0$, with probability at least $1-\delta$ we have the following holds:*

$$d(\tilde{P}_{trained}(y_i|\boldsymbol{x}_i), P^{true}(y_i|\boldsymbol{x}_i)) \le \varepsilon_{noise} + \lambda_1\Re(\mathcal{F}) + \lambda_2 l_\infty^{noisy}\sqrt{\frac{\log 1/\delta}{N}}$$

*with $\lambda_1, \lambda_2 > 0$. Here, $\varepsilon_{noise}$ denotes the difference term induced by the distribution difference between $P$ and $P^{true}$.*

With theorem 1 and theorem 2, ignoring the uncontrollable and common optimization bound error terms (marked in gray), we confirm that the zero-shot classifier estimation is highly related to domain gap and prompts quality while the trained classifier estimation is affected by the noise of in-question dataset, which is intuitively consistent with our expectation. Moreover, we empirically verify that the higher the noise ratio, the greater the performance advantage of zero-shot classifier over the trained classifier (section 4.1). More importantly, $\varepsilon_{noise}$ is always inevitable while $\Delta$ can be easily improved with better prompt engineering and $\varepsilon_{domain}$ can be also reduced by training CLIP with more abundant dataset and thus minimizing the domain gap.

**Proof**   To start with, we first state the essential generalization error bound based on Rademacher complexity ($\mathfrak{R}$):

**Lemma 1** (Rademacher generalization error bound (Mohri et al., 2018)). *Supposing we have $N$ i.i.d samples $\{x_i\}_{i=1}^N$ from distribution $P(x)$. Let $\mathcal{F}$ be the hypothesis space of model $f$ and $L$ be any classification-calibrated surrogate loss function of 0-1 loss ranging from $[a,b]$. Then, for any $\delta > 0$, with probability at least $1 - \delta$ we have the following holds for all $f \in \mathcal{F}$:*

$$R(f) \leq \hat{R}(f) + 2\mathfrak{R}(\mathcal{F}) + (b-a)\sqrt{\frac{\log(1/\delta)}{2N}}$$

*Here, $R^P(f) = E_{P(x)}L(x; f)$ denotes the expected risk with $f$ and $\hat{R}^P(f) = \frac{1}{N}\sum_{i=1}^N L(x_i; f)$ denotes the empirical one. Please do not confuse the notations here with other notations.*

### G.1   DERIVATION OF THEOREM 1

Let us recall the formulation of CLIP model. CLIP aims to learn from a dataset of image-text pairs, denoted as $(x_i, z_i)_{i=1}^M$, which is *i.i.d.* sampled from a hidden joint distribution $Q(x, z)$ with $\sup(Q) = \{x \in \mathbb{R}^{C \times H \times W}, z \in \mathbb{R}^d\}$. As the dataset for training CLIP is often also considered 'noisy' [4]. Here, we denote the clean joint distribution for CLIP training dataset as $Q'(x, z)$ and the corresponding clean dataset as $(x_i, z_i')_{i=1}^M$.

According to eq. (3) in main paper, to measure the distance between $\tilde{P}_{zeroshot}(y|x)$ with $P^{true}(y|x)$, we then divide it into two parts, i.e, the distance between $\tilde{P}_{zeroshot}(y|x)$ and $Q'(y|x)$ (*Model error*) and the distance between $Q'(y|x)$ and $P^{true}(y|x)$ (*Domain gap*).

On the one hand, we simply define the *domain gap* as $\varepsilon_{domain}$ here, which represents how different the true prediction distribution ($Q'(y|x)$) of CLIP training dataset is than the true prediction distribution ($P^{true}(y|x)$) of out targeted classification problem. This is technically irreducible but can be improved by making the CLIP training dataset more abundant and reduce its domain gap with the targeted classification dataset.

On the other hand, the model error is further divided into two parts:

1. the distance between $Q_{g,h}(z|x)$ and $Q'(z|x)$ (*CLIP generalization error*);

2. the error induced by eq. (4) when estimating $\tilde{P}_{zeroshot}(y|x)$ based on $Q'(x, z)$ (*Prompt sampling and designing*).

Intuitively, the first part represents how good our CLIP model learn and generalize, and the second part represents how much extra bias we introduce when we try to approximate the integral with sampling (eq. (4)).

**CLIP generalization error**   Following main paper's notations, let us recall here the empirical risk on *i.i.d* sampled dataset from the noisy CLIP distribution $Q$ as $\hat{R}^Q(f)$:

$$\hat{R}^Q(g, h) = \frac{1}{M}\sum_{i=1}^M L_{clip}(x_i, z_i; g, h),$$

and the corresponding empirical risk *w.r.t* clean dataset as:

$$\hat{R}^{Q'}(g, h) = \frac{1}{M}\sum_{i=1}^M L_{clip}(x_i, z_i'; g, h),$$

while the expected risk on the unknown clean CLIP distribution $Q'$ as $R^{Q'}(g, h)$, as:

$$R^{Q'}(g, h) = E_{Q'}L_{clip}(x, z; g, h)$$

---

[4]The image description sometimes can be random due to the data collection process (Jia et al., 2021; Radford et al., 2021). We here also consider this into consideration. **Please note this is different with our interested label noise in this work**.

Below we present how to bound the CLIP generalization error. We denote $(\hat{g}, \hat{h}) = \arg\min_{g \in \mathcal{G}, h \in \mathcal{H}} \hat{R}^Q(g, h)$ as the empirical optimal model *w.r.t i.i.d* sampled dataset from $Q$, $(g^*, h^*) = \arg\min_{g \in \mathcal{G}, h \in \mathcal{H}} R^{Q'}(g, h)$ as the best-achievable model *w.r.t* clean distribution $Q'$ and $(g_{bayes}, h_{bayes}) = \arg\min_{g, h} R^{Q'}(g, h)$ as the Bayes optimal model *w.r.t* clean distribution $Q'$. We can decompose the excess risk of our learned empirical optimal model $\hat{f}$ over the Bayes optimal model $f_{bayes}$ as:

$$
\begin{aligned}
R^{Q'}(\hat{g}, \hat{h}) - R^{Q'}(g_{bayes}, h_{bayes}) &= \underbrace{R^{Q'}(\hat{g}, \hat{h}) - R^{Q'}(g^*, h^*)}_{estimation\ error} + \underbrace{R^{Q'}(g^*, h^*) - R^{Q'}(g_{bayes}, h_{bayes})}_{approximation\ error} \\
&= R^{Q'}(\hat{g}, \hat{h}) - R^{Q'}(g^*, h^*) + \mathcal{B}_{approx} \\
&\approx R^{Q'}(\hat{g}, \hat{h}) - R^{Q'}(g^*, h^*)
\end{aligned}
\tag{9}
$$

Exact analysis of approximation error is often intractable, we thus abbreviate it as $\mathcal{B}_{approx}$ and omit it in subsequent analysis. For estimation error, we have:

$$
\begin{aligned}
R^{Q'}(\hat{g}, \hat{h}) - R^{Q'}(g^*, h^*) &= R^{Q'}(\hat{g}, \hat{h}) - \hat{R}^Q(\hat{g}, \hat{h}) + \hat{R}^Q(\hat{g}, \hat{h}) \\
&\quad - \hat{R}^Q(g^*, h^*) + \hat{R}^Q(g^*, h^*) - R^{Q'}(g^*, h^*) \\
&\xrightarrow{\hat{R}^Q(\hat{g}, \hat{h}) - \hat{R}^Q(g^*, h^*) \leq 0} \\
&\leq R^{Q'}(\hat{g}, \hat{h}) - \hat{R}^Q(\hat{g}, \hat{h}) + \hat{R}^Q(g^*, h^*) - R^{Q'}(g^*, h^*) \\
&\leq 2\sup_{g \in \mathcal{G}, h \in \mathcal{H}} |R^{Q'}(g, h) - \hat{R}^Q(g, h)|
\end{aligned}
\tag{10}
$$

Supposing the range of $L_{clip}$ as $[0, l_\infty^{clip}]$ for all $(\boldsymbol{x}, \boldsymbol{z})$ in $\sup(Q)$ with $g, h \in \mathcal{G}, \mathcal{H}$ and $L_{clip}$ is $\lambda$-Lipschitz continuous w.r.t $\boldsymbol{z}_i$, according to Lemma 1 and triangle inequality, we have:

$$
\begin{aligned}
|R^{Q'}(g, h) - \hat{R}^Q(g, h)| &\leq \overbrace{|R^{Q'}(g, h) - \hat{R}^{Q'}(g, h)|}^{Lemma\ 1} + \overbrace{|\hat{R}^{Q'}(g, h) - \hat{R}^Q(g, h)|}^{Lipschitz\ continuous} \\
&\leq 2\mathfrak{R}(\mathcal{G} \circ \mathcal{H}) + l_\infty^{clip} \sqrt{\frac{\log(1/\delta)}{2M}} + \lambda \frac{1}{M} \sum_{i=1}^{M} \|\boldsymbol{z}_i - \boldsymbol{z}_i'\|_2 \\
&\leq 2\mathfrak{R}(\mathcal{G} \circ \mathcal{H}) + l_\infty^{clip} \sqrt{\frac{\log(1/\delta)}{2M}} + \varepsilon_n
\end{aligned}
\tag{11}
$$

Here, we rewrite $\lambda \frac{1}{M} \sum_{i=1}^{M} \|\boldsymbol{z}_i - \boldsymbol{z}_i'\|_2$ as $\varepsilon_n$ which is the error term induced by language noise ($\boldsymbol{z}_i \neq \boldsymbol{z}_i'$). With eq. (9) and eq. (11), we have:

$$
R^{Q'}(\hat{g}, \hat{h}) - R^{Q'}(g_{bayes}, h_{bayes}) \leq 2(2\mathfrak{R}(\mathcal{G} \circ \mathcal{H}) + l_\infty^{clip} \sqrt{\frac{\log(1/\delta)}{2M}} + \varepsilon_n)
\tag{12}
$$

To further connect the generalization error bound above and the distance of estimated probability $Q_{g,h}(z|x)$ and $Q'(z|x)$, we have:

$$
\begin{aligned}
R^{Q'}(g, h) &= E_{Q'} L_{clip}(x, z; g, h) \\
&= -\frac{1}{2} \int Q'(x) \int Q'(z|x) \log Q_{g,h}(z|x) dz dx \\
&\quad -\frac{1}{2} \int Q'(z) \int Q'(x|z) \log Q_{g,h}(x|z) dx dz \\
&= \frac{1}{2} \int Q'(x) D_{KL}(Q'(z|x), Q_{g,h}(z|x)) dx - \frac{1}{2} \int Q'(x) \int Q'(z|x) \log Q'(z|x) dz dx \\
&\quad -\frac{1}{2} \int Q'(z) \int Q'(x|z) \log Q_{g,h}(x|z) dx dz \\
&\geq \frac{1}{2} \int Q'(x) D_{KL}(Q'(z|x), Q_{g,h}(z|x)) dx \\
&\geq d(Q_{g,h}(z|x), Q'(z|x))
\end{aligned}
\tag{13}
$$

Specifically, we have $(g_{bayes}, h_{bayes}) = \arg\min R^Q(g, h)$ when and only when $Q_{g,h}(z|x) = Q'(z|x)$. Intuitively, when and only when the learned model is Bayes optimal, we have a zero distance between the estimated probability and the ground-truth probability. According to eq. (12), we thus have:

$$
R^{Q'}(\hat{g}, \hat{h}) - R^{Q'}(g_{bayes}, h_{bayes}) \leq 2(2\Re(\mathcal{G} \circ \mathcal{H}) + l_\infty^{clip} \sqrt{\frac{\log(1/\delta)}{2M}} + \varepsilon_n) \implies
$$

$$
d(Q_{\hat{g}, \hat{h}}(z|x), Q'(z|x)) \leq R^{Q'}(g_{bayes}, h_{bayes}) + 2(2\Re(\mathcal{G} \circ \mathcal{H}) + l_\infty^{clip} \sqrt{\frac{\log(1/\delta)}{2M}} + \varepsilon_n)
$$

$$
\leq 2(2\Re(\mathcal{G} \circ \mathcal{H}) + l_\infty^{clip} \sqrt{\frac{\log(1/\delta)}{2M}} + \varepsilon_n)
\tag{14}
$$

**Prompt sampling and designing** We then take step two into consideration. According to eq. (3), with $Q_{\hat{g}, \hat{h}}(z|x)$ we can estimate $\tilde{P}_{zeroshot}(y|x)$. To quantify the additional error of the sampling process (eq. (4)), we denote as $\Delta$ a error coefficient which represents how much extra error been induced. Let us recall the domain gap ($\varepsilon_{domain}$) before, we thus have Theorem 1 below:

**Theorem** (ESTIMATION WITH ZERO-SHOT CLASSIFIER). *Let $\mathcal{G}, \mathcal{H}$ be the hypothesis space of vision encoder $g$ and language encoder $h$. Let us denote the rademacher complexity as $\Re(\mathcal{G} \circ \mathcal{H})$ of the combined CLIP model. Supposing the range of $L$ from eq. (2) as $[0, l_\infty^{clip}]$ for all $(x, z)$ in $\sup(Q)$ with $g, h \in \mathcal{G}, \mathcal{H}$. Then, for any $\delta > 0$, with probability at least $1 - \delta$ we have the following holds:*

$$
d(\tilde{P}_{zeroshot}(y_i|x_i), P^{true}(y_i|x_i)) \leq \varepsilon_{domain} + \Delta(\lambda_1 \Re(\mathcal{G} \circ \mathcal{H}) + \lambda_2 l_\infty^{clip} \sqrt{\frac{\log 1/\delta}{M}} + \lambda_3 \varepsilon_n)
$$

*with $\lambda_1, \lambda_2, \lambda_3 > 0$. Here, $\varepsilon_{domain}$ denotes the bias term induced by the domain gap between $Q$ and $P^{true}$, and $\Delta \geq 1$ denotes the bias coefficient induced in designing prompts and sampling in eq. (3).*

## G.2 DERIVATION OF THEOREM 2

The derivation of Theorem 2 follows a similar but rather simpler process. Specifically, with $Q, Q', z_i, z_i', M$ replaced by $P, P', y_i, y_i', N$, similar to eq. (12), we have:

$$
R^{true}(\hat{f}) - R^{true}(f_{bayes}) \leq 2(2\Re(\mathcal{F}) + l_\infty^{noisy} \sqrt{\frac{\log(1/\delta)}{2N}} + \varepsilon_{noise})
\tag{15}
$$

To similarly connect the generalization error bound above and the distance of estimated probability $P_f(y|\boldsymbol{x})$ and $P^{true}(y|\boldsymbol{x})$, with $L_{noisy}$ as the cross-entropy loss, we have:

$$
\begin{aligned}
R^{true}(f) &= E_{P^{true}} L_{noisy}(\boldsymbol{x}, y; f) \\
&= -\int P^{true}(\boldsymbol{x}) \int P^{true}(y|\boldsymbol{x}) \log P_f(y|\boldsymbol{x}) dy d\boldsymbol{x} \\
&= \int P^{true}(\boldsymbol{x}) D_{KL}(P^{true}(y|\boldsymbol{x}), P_f(y|\boldsymbol{x})) d\boldsymbol{x} \\
&\quad - \int P^{true}(\boldsymbol{x}) \int P^{true}(y|\boldsymbol{x}) \log P^{true}(y|\boldsymbol{x}) dy d\boldsymbol{x} \\
&\geq \int P^{true}(\boldsymbol{x}) D_{KL}(P^{true}(y|\boldsymbol{x}), P_f(y|\boldsymbol{x})) d\boldsymbol{x} \\
&\geq 2d(P_f(y|\boldsymbol{x}), P^{true}(y|\boldsymbol{x}))
\end{aligned}
\tag{16}
$$

Similarly, we then have Theorem 2:

**Theorem** (ESTIMATION WITH TRAINED CLASSIFIER ). *Let $\mathcal{F}$ be the hypothesis space of trained classifier $f'$. Let us denote the rademacher complexity as $\mathfrak{R}(\mathcal{F})$ of the trained classifier. Supposing the range of $L$ for training $f'$ as $[0, l_\infty^{noisy}]$ for all $(\boldsymbol{x}, y)$ in $\sup(P)$ with $f' \in \mathcal{F}$. Then, for any $\delta > 0$, with probability at least $1 - \delta$ we have the following holds:*

$$
d(\tilde{P}_{trained}(y_i|\boldsymbol{x}_i), P^{true}(y_i|\boldsymbol{x}_i)) \leq \varepsilon_{noise} + \lambda_1 \mathfrak{R}(\mathcal{F}) + \lambda_2 l_\infty^{noisy} \sqrt{\frac{\log 1/\delta}{N}}
$$

*with $\lambda_1, \lambda_2 > 0$. Here, $\varepsilon_{noise}$ denotes the difference term induced by the distribution difference between $P$ and $P^{true}$.*

