# OpenReview forum: "Cleaning label noise with vision-language models"
_ICLR.cc/2024/Conference — Submitted to ICLR 2024_

### Official Review · Reviewer_tEQg · 2023-10-31

**Soundness:** 3 good
**Presentation:** 3 good
**Contribution:** 3 good
**Rating:** 5
**Confidence:** 5

**Summary:**

This paper focuses on the integration of pretrained vision-language models, like CLIP, into the process of learning from noisy labels. To this end, the authors introduce a method called CLIPSelector, which leverages CLIP's powerful zero-shot classifier and an easily-inducible classifier based on CLIP's vision encoder to select clean samples. Additionally, they introduce a semi-supervised learning approach called MixFix to gradually incorporate missing clean samples and re-label noisy samples based on varying thresholds to enhance performance. The authors validate their approach through a series of experiments on different benchmarks, including datasets with synthetic and real-world noise.

**Strengths:**

1. This paper breaks new ground by exploring the use of pretrained vision-language models, such as CLIP, to address the challenge of noisy labels. This approach is promising as it goes beyond relying solely on information from the noisy dataset.
2. The fixed hyperparameters across all experiments showcase the robustness and practicality of the proposed method.

**Weaknesses:**

1. My major concern is the potential unfair comparisons. The notable performance improvements shown in Tables 2-4 could be attributed to CLIP's superior representation learning capabilities. A fairer comparison could involve replacing the baselines' backbone with CLIP's visual encoder. Furthermore, Table 3 lacks comparison results with recent works focused on instance-dependent noise from 2022-2023.
2. Discrepancies between the CLIP's zero-shot results on CIFAR in Table 1 and the original paper need clarification.
3. The claims regarding inferior performance on Red Mini-ImageNet require more explanation and context.
4. What does SOTA in Table 1 means? Please supplement the necessary details.
5. Ambiguous statements like "on the clean test set of in-question noisy datasets" should be elucidated to enhance clarity.
6. The derivation of Eq. 4 from Eq. 3 is not explained, and the effect of the added class feature in the prompt remains unclear. Additional ablation studies are necessary to substantiate these claims.

**Questions:**

Please see the weaknesses, thx

---

> ### Author Response · Authors · 2023-11-13
> **Reply to Reviewer tEQg (part 1)**
>
> Thanks for the insightful reviews and acknowledgement about the method and presentation. We are happy to answer your raised questions below:
> > *Q1.1.My major concern is the potential unfair comparisons. The notable performance improvements shown in Tables 2-4 could be attributed to CLIP's superior representation learning capabilities. A fairer comparison could involve replacing the baselines' backbone with CLIP's visual encoder.*
>
> We understand the reviewers' concerns, but we chose the specific settings precisely to allow for fair comparisons: as it stands, all our comparisons use the same backboneas reported in the original paper. Our main motivation is to bring the benefits of using visual language models to the "Learning with Noisy Labels" community -- we believe that this is a significant contribution. We show that with a very simple solution we achieve SOTA. There are indeed many other ways that VL models could be used in this domain, we simply use CLIP for sample selection to avoid its direct impact on the in-training model to further ensure a fair comparison. And we hope that other researchers will explore them in their work. See our response to reviewer s2nG Q2 for more discussion and results.
>
> > *Q1.2.A fairer comparison could involve replacing the baselines' backbone with CLIP's visual encoder.*
>
> Please refer to our response to reviewer Vnif Q1.2.
>
> >*Q1.3. Furthermore, Table 3 lacks comparison results with recent works focused on instance-dependent noise from 2022-2023.*
>
> We will incorporate more recent results in Table 3 in the updated verision. Due to time limitation, we present below an initial version in Table R2 with one more recent work for your perusal:
> | Method | 10\% | 20\% | 30\% | 40\% |
> |---|---|---|---|---|
> | Cross-Entropy | 91.25 | 86.34 | 80.87 | 75.68 |
> | F-correction | 91.06 | 86.35 | 78.87 | 71.12 |
> | Co-teaching | 91.22 | 87.28 | 84.33 | 78.72 |
> | GCE | 90.97 | 86.44 | 81.54 | 76.71 |
> | DAC | 90.94 | 86.16 | 80.88 | 74.80 |
> | DMI | 91.26 | 86.57 | 81.98 | 77.81 |
> | SEAL | 91.32 | 87.79 | 85.30 | 82.98 |
> | SELC [1] | 91.63 | 88.33 | 86.28 | 84.23 |
> | Cross-Entropy (reproduced) | 90.76 | 86.08 | 80.64 | 75.27 |
> | CLIPSelector + Cross-Entropy | **92.33** | **91.06** | **89.71** | **88.26** |
>
> Table R2. More results on instance-dependent noise from [2]
>
> We are happy to include more discussions if the reviewer is aware of any other related works reported on the same bencmarks. There are also other works proposing different models for instance-dependent noise such as [2]. We here report extra results in Table R3 with noisy labels provided by [3] (https://github.com/pxiangwu/PLC/tree/master/cifar/noisy_labels). According to Tables R2 and R3, we can verify that at various noise ratios and noise types, significant improvements are obtained by simply introducing CLIP-based sample selection.
>
> | Method | Type-I (35%) | Type-I (70%) | Type-II (35%) | Type-II (70%) | Type-III (35%) | Type-III (70%) |
> |---|---|---|---|---|---|---|
> | Co-teaching+ | 79.97±0.15 | 40.69±1.99 | 77.35±0.44 | 45.44±0.64 | 78.38±0.67 | 41.90±0.86 |
> | GCE | 80.65±0.39 | 36.52±1.62 | 77.60±0.88 | 40.30±1.46 | 79.18±0.61 | 37.10±0.59 |
> | SL | 79.76±0.72 | 36.29±0.66 | 77.92±0.89 | 41.11±1.92 | 78.81±0.29 | 38.49±1.46 |
> | LRT | 80.98±0.80 | 41.52±4.53 | 80.74±0.25 | 44.67±3.89 | 81.08±0.35 | 44.47±1.23 |
> | PLC [3] | 82.80 ± 0.27 | 42.74 ± 2.14 | 81.54 ± 0.47 | 46.04 ± 2.20 | 81.50 ± 0.50 | 45.05 ± 1.13 |
> | MDDC     [4] | 83.12±0.44 | 43.70±0.91 | 80.59±0.19 | 47.06±0.41 | 82.02±0.23 | 46.31±0.92 |
> | CDDC  [4] | 83.17±0.43 | 45.14±1.11 | 80.76±0.25 | 48.10±1.74 | 81.58±0.07 | 47.92±1.28 |
> | CLIPSelector + Cross-Entropy | **89.35±0.24** | **67.47±0.56** | **88.59±0.27** | **67.74±0.46** | **89.60±0.29** | **66.84±0.87** |
>
>
> Table R3. Results on instance-dependent noise from [3]
>
> > *Q2.Discrepancies between the CLIP's zero-shot results on CIFAR in Table 1 and the original paper need clarification.*
>
> Admittedly, reproducing the reported results from the original CLIP paper is actually challenging, as acknowledged in the discussion here: https://github.com/openai/CLIP/issues/164.  The reported numbers in our paper were obtained under consistent experimental settings to ensure a fair and reliable comparison.

---

> > ### Author Response · Authors · 2023-11-13
> > **Reply to Reviewer tEQg (part 2)**
> >
> > > *Q3.The claims regarding inferior performance on Red Mini-ImageNet require more explanation and context.*
> >
> > Thank you for the careful review. Here, we briefly outline the data collection process for the Red Mini-ImageNet dataset to provide some insights [2]. Red Mini-ImageNet is constructed by substituting the original samples in the Mini-ImageNet dataset with collected noisy samples at specific ratios. The noisy samples for replacement are obtained by crawling using search engines and category names as keywords. Consequently, we conjecture that this process introduces a small domain gap between Red Mni-ImageNet dataset and the larg-scale Web dataset used for training CLIP, contributing to the positive performance in zero-shot classification. Regarding the comparatively lower performance of our method and the state-of-the-art (SOTA) here, we attribute  it to the use of a relatively small-capacity model (PreResNet18 \~11m parameters) in comparison to the CLIP backbone (VIT-B/32 \~86m parameters).
> >
> >
> > > *Q4.What does SOTA in Table 1 means? Please supplement the necessary details.*
> >
> > Here ‘SOTA’ represents the best results summarized from subsequent Tables 2-4 in the paper. In Table 1,  we tend to emphasize a more direct comparison of SOTA vs Ours VS CLIP zero-shot classification.
> >
> > > *Q5.Ambiguous statements like "on the clean test set of in-question noisy datasets" should be elucidated to enhance clarity.*
> >
> > We apologize for any confusion. To clarify, when we mention ‘the clean test set of in-question noisy datasets’, we mean the clean test set corresponding to the noisy train set. We will address and rectify this point in the updated version of the paper.
> >
> >
> > > *Q6.The derivation of Eq. 4 from Eq. 3 is not explained, and the effect of the added class feature in the prompt remains unclear. Additional ablation studies are necessary to substantiate these claims.*
> >
> > We apologize for any confusion. The transition from eq.3 to eq.4 is primarily based on the assumption that the designed prompts exclusively correspond to specific semantic classes ($Q(y=y_i|z=`\texttt{A photo of {class name of $y_i$}.}') \approx 1$). We will provide additional details about these equations. For a comprehensive analysis of different prompts and their impact, please refer to Table 9 in Appendix C of the original paper. In that table, we compare various prompting styles for zero-shot classification, which serves as the surrogate task for estimating the probability using a zero-shot classifier (our first method for estimating the probability).
> >
> > If the reviewer has any additional questions or anything else to discuss, we are more than happy to engage further in the conversation.
> >
> > *[1] SELC: Self-Ensemble Label Correction Improves Learning with Noisy Labels , IJCAI2022.*
> >
> > *[2] Beyond class-conditional assumption: A primary attempt to combat instance-dependent label noise, AAAI2021.*
> >
> > *[3] Learning with Feature-Dependent Label Noise: A Progressive Approach, ICLR2021.*
> >
> > *[4] Tackling Instance-Dependent Label Noise with Dynamic Distribution Calibration, ACMMM2022.*

---

> > > ### Comment · Reviewer_tEQg · 2023-11-14
> > > **Reply to ``Reply to Reviewer tEQg (part 1&2)"**
> > >
> > > I commend the authors for their comprehensive response, including detailed statements and additional experimental results. However, a significant concern, echoing the sentiment of Reviewer Vnif, remains unaddressed - the replacement of the baselines' backbone with CLIP's visual encoder. This issue is crucial for the thorough understanding and validation of the proposed approach.
> > >
> > > Furthermore, I share the sentiment expressed by Reviewer GWvS regarding the paper's clarity. The confusion noted in the writing style, along with the perceived unnecessary complexity in the presentation of deviation details, hinders the paper's accessibility for readers. Clarifying and simplifying these aspects would greatly enhance the overall quality of the manuscript.
> > >
> > > Regrettably, in light of these concerns, I maintain my negative rating for the submission.

---

> ### Author Response · Authors · 2023-11-14
> **Further reply to Reviewer tEQg**
>
> We appreciate your prompt reply. We would like to further clarify your concerns below:
> >*Q7: Backbone Replacement with CLIP's Visual Encoder.*
>
> To finetune the CLIP visual encoder is another available style utilizing a vision-language model for LNL. Instead, we use offline sample selection, specifically because:
>
> - **Acknowledgement**: sample selection is the dominant paradigm for LNL, by utilizing CLIP for sample selection our method can be seamlessly utilized with existing methods. For example, simply adding an extra warmup stage before the original training frmework with the selected samples from CLIP.
>
> - **Effectiveness**: Viewing CLIP as a 'black-box foundation model', considering the possible computational cost/deployment issues, etc, the most direct and convenient way to utilize it would be the offline one.
>
> We expect our method to **motivate the usage of CLIP in the LNL community since CLIP with only naive/offline sample selection shows competitive results**. We hope that other researchers in the field of LNL will explore other styles using vision-language methods in their work.
>
>
> **Fine-tuning experiment:** Regarding the suggestion to replace the model with CLIP's pretrained model, we here report fine-tuning results on SSR[1], as mentioned by Reviewer Vnif. Specifically, we fine-tune the CLIP pre-trained model with VIT-B/32 backbone, using its visual encoder with an additional linear layer classifier. We initially froze the encoder, train the classifier for one epoch with LR=0.02, and subsequently train the whole model with LR=0.001 and LR_scheduler=CosineAnnealing for 100 epochs, with a weight decay of 0.01. As a comparison, we use the same structure but randomized the weights of the encoder, train for 100 epochs with LR=0.001 and LR_scheduler=CosineAnnealing, with a weight decay of 0.01.  Initial results are presented below in Table R4:
>
> | Method | CIFAR10 90% symmetric noise |
> |---|---|
> | SSR | 93.45 |
> | CLIP+SSR | 95.98 |
>
> Table R4. SSR with CLIP
>
> As expected, the pre-trained CLIP encoder provides effective improvement, further demonstrating the potential of the CLIP model for LNL. Also, please note that fine-tuning CLIP is not trivial, as described in the following GitHub issues. Through better hyperparameter settings, we believe there is still room for further progress.
> - https://github.com/openai/CLIP/issues/83
> - https://github.com/openai/CLIP/issues/150
> - https://github.com/openai/CLIP/issues/362
>
> ---
>
> >*Q8: Paper's Clarity.*
>
> Regarding the presentation of the paper, we are currently making revisions to the manuscript based on the feedback from all reviewers. We would like to convey that the majority of the changes are straightforward and actionable.
>
> We hope that this further clarification addresses your concerns. If you have any additional questions, we are open to further discussion.
>
> *[1] SSR: An Efficient and Robust Framework for Learning with Unknown Label Noise, BMVC2022.*

---

> > ### Comment · Reviewer_tEQg · 2023-11-22
> > **Further concerns**
> >
> > Thank you for the additional clarification and conducting further experiments. Upon reviewing Table R4, I observed that the SSR results (93.45) are lower than those reported in [1] (95.2). I'd also like to confirm: does 'SSR' in Table R4 refer to SSR utilizing CLIP as the backbone, while 'CLIP+SSR' denotes SSR employing the suggested CLIP selector?

---

> > > ### Author Response · Authors · 2023-11-22
> > > **Further reply**
> > >
> > > We appreciate your prompt reply. We would like to further clarify your concerns briefly. Due to the differences in model architecture and length of training epochs, the numbers here are not for comparison with the results reported in the original paper. Instead, our goal is to ablate and validate the benefits of the CLIP pre-trained encoder, as suggested. Specifically, "CLIP+SSR" represents the use of the CLIP pre-trained encoder with the SSR method, while "SSR" represents the same structure but with randomly initialized encoder.

---

> > > > ### Comment · Reviewer_tEQg · 2023-11-22
> > > >
> > > > Thank you for your prompt response. However, my concern remains unresolved regarding Table R4. Specifically, I believe that comparing the proposed method to CLIP+SSR is a fair assessment. Yet, the performance of CLIP+SSR (95.98) surpasses that of your approach (94.23). It would be beneficial to include the results of 'SSR+your approach' to thoroughly explore the effectiveness of the proposed method.

---

> ### Author Response · Authors · 2023-11-22
> **Further reply**
>
> We appreciate your prompt reply and willingness to engage in further discussion. However, we would like to respectfully reiterate that comparing the online usage of large-scale pretrained CLIP models to offline sample selection of CLIP falls outside the scope of this work -  as discussed in the paper, the best style for CLIP in LNL is still underexplored. Besides, It's a well-established fact that directly finetuning large-scale pretrained models is highly possible to yield superior performance compared to offline usage.
> **Our primary goal is to introduce CLIP to the LNL community, and we believe this comparison is unnecessary and distracts from our focus.** We kindly refer the reviewer's attention to the **<Additional comments about the fairness and contribution of the proposed method>** response section for a more in-depth explanation of our approach's significance. Thanks for your time and consideration.
>
> **Specifically, regarding the numbers in Table R4, we can actually never make a comparison. For example, on CIFAR dataset, CLIP model applies VIT-B/32 with 224x224 px image input, while all current works utilize PreResNet18 with 32x32 px input. This is why we insist a offline usage of CLIP model to avoid effects of these extra factors in the original paper.**

---

> > ### Comment · Reviewer_tEQg · 2023-11-22
> >
> > Thank you for your quick response. While I understand your points, I remain concerned that comparing the online usage of CLIP models versus the proposed offline sample selection of CLIP would offer a more compelling comparison.

---

> > > ### Author Response · Authors · 2023-11-22
> > > **Further reply**
> > >
> > > We sincerely appreciate your understanding and response. This helps us better understand how more researchers view the potential of CLIP in LNL. In the updated manuscript, we include preliminary experimental results about online usage in the appendix. We will conduct and include more experiments with CLIP if we get further manuscript updating opportunity. Thanks very much for your consideration.

---

### Official Review · Reviewer_Vnif · 2023-10-31

**Soundness:** 3 good
**Presentation:** 3 good
**Contribution:** 2 fair
**Rating:** 5
**Confidence:** 4

**Summary:**

This paper proposes using the pre-trained vision-language model CLIP for sample selection to mitigate selfconfirmation
bias. Specifically, they introduce the CLIPSelector, which utilizes both the CLIP’s zero-shot
classifier and an easily-inducible classifier based on its vision encoder and noisy labels for sample selection.
And they further introduce a semi-supervised learning method called MixFix.

**Strengths:**

1. The paper is well presented and explains the algorithm and experiments clearly.
2. The experiments are conducted on various datasets.

**Weaknesses:**

1. The performance lacks some competitiveness. Some methods are not compared, for example, SSR: An
Efficient and Robust Framework for Learning with Unknown Label Noise.
2. The main idea of the paper is to use the CLIP zero-shot classifier for sample selection and lacks novelty. And
the semi-supervised learning methods has also been applied in previous works.

**Questions:**

1. This paper uses the CLIP pre-trained model, I think this is unfair for previous works without pre-trained model. Combining
previous methods with training from CLIP pre-trained model other than training from scratch should also be compared.
2. Equations (2) misses ).

---

> ### Author Response · Authors · 2023-11-13
> **Reply to Reviewer Vnif**
>
> Thanks for the insightful reviews and acknowledgement about the experiments and presentation. We are happy to answer your raised questions below:
>
> > *Q1.1.This paper uses the CLIP pre-trained model, I think this is unfair for previous works without pre-trained model. *
>
> We understand the concerns of the reviewer but we chose the specific setting precisely so as to perform fair comparisons: as it stands, all our comparisons are using the same encoders (e.g. PreResNet18) as reported in the original papers. We note that , in several cases,  CLIP-based encoders may be heavy and not desired to be used. In this condition, sample selection enables offline and efficient usage of CLIP model and ensures a fair comparison for all methods.
>
> Our main motivation is to introduce the benefits of using vision-language models into the “Learning with Noisy Labels” community -- we believe that this is a significant contribution. We show that with a very simple scheme we achieve SOTA. There are indeed many other ways that VL models could be used in this domain, we simply use CLIP for sample selection to avoid its direct impact on the in-training model to further ensure a fair comparison.  We hope that other researchers in the field of LNL will explore using VL methods in their work. Please refer to our reply to Q2 of Reviewer s2nG for more discussions and results.
>
> > *Q1.2.Combining previous methods with training from CLIP pre-trained model other than training from scratch should also be compared.*
>
> Integrating CLIP with existing techniques for learning with noisy labels is an interesting research direction that we intend to explore in the future. Preliminary experiments demonstrate that a pretrained clip backbone does helps. In this paper, however, we focus on utilising CLIP as a sample selection method that does not require any adaptation to the training of the classifier,  and achieve SOTA/competitive results. We believe that this is a sufficient contribution.
>
> > *Q2.Equations (2) misses ).*
>
> We apologize for the typo, we are fixing this along with other changes. We will upload the updated verison later.
>
> If the reviewer has any additional questions or anything else to discuss, we are more than happy to engage further in the conversation.
>
> *[1] SSR: An Efficient and Robust Framework for Learning with Unknown Label Noise.*

---

> > ### Author Response · Authors · 2023-11-22
> > **Further reply**
> >
> > Thank you very much for your time and effort in reading and improving our paper. As the discussion period is coming to an end, we would like to know if you have any further questions.
> >
> > In addition, regarding *Q1.2*, we would like to draw your attention to our further reply for reviewer tEQg. We have included the experiments you suggested and verified the effectiveness of fine-tuning the CLIP pre-trained encoder.
> >
> > Thank you again for your time and consideration.

---

### Official Review · Reviewer_GWvS · 2023-11-08

**Soundness:** 2 fair
**Presentation:** 1 poor
**Contribution:** 3 good
**Rating:** 3
**Confidence:** 3

**Summary:**

The paper addresses the issue of learning with noisy labels. They present an approach based on learning to select samples from a downstream dataset optimally to improve performance for a downstream task. Their approach is based on the CLIP model and is named CLIPSelector. Their central idea is to use a thresholding mechanism based on the zero-shot ability of the CLIP model to enable selection of cleaner samples and to detect which samples need to be relabeled. They utilize this approach to data augment the training set gradually thereby increasing sample difficulty by using the predictions of the trained model.

**Strengths:**

Overall the paper addresses an important problem of learning under noise labels, which is critical for ML deployment. Moreover, the authors use an auxiliary foundation model that enables sample selection and since their approach is modular, this model can be substituted for a stronger model in the future. The hyper-parameter experiments for different thresholds will be useful for the readers. Additional discussions about the applicability of the method and how it performs on granularly labeled dataset (including identifying its shortcomings) is very welcome.

**Weaknesses:**

W1: The biggest weakness of the paper is the writing. the authors have made the paper extremely complicated with inconsistent and complex notation. For eg: the addition of theorems 1 and 2 is not necessary for the paper, they can be relegated to the appendix. In addition, the strength of the inequality relies on the tightness of the bound. So it isn't a surprise that the conclusions drawn from the theorems hold, but the key point is how tightly they hold, which is impossible to know. Several important details that are required to be in the paper are relegated to the appendix, such as the hyper-parameter ablations on theta_r. Overall, the approach can be explained more simply and clearly instead of the complex framework that the authors have presented here, which seems unnecessary.

W2: Incomplete description of experimental setups. The experiments section does not appear well constructed, although the experiments themselves are useful. For instance, it is unclear why Sec 4.1 exists before the results about model performance. The explanation of the first paragraph of section 1 is incredibly hard to parse through.

W3: No qualitative results are presented. The authors present results on traditional benchmarks and claim their method performs better than SOTA (it isn't clear what SOTA is here from the tables), but fail to ask the question why do their approach perform better? What is the difference in behavior between "easy noise" and "hard noise"? Absence of qualitative analysis make it a subpar presentation for the reader.

**Questions:**

1. Eq1: Sample selection mechanism takes as input the predicted probability and the label? Please clarify.
2. Typo: Sec 3.2: “we consistent the notations for CLIP’s training”
3. Clarify Sec 4.1 “clean test set of in-question noisy datasets”
4. Appendix F: "label noise normally does not affect the sample itself": Label noise can be highly sample dependent, so I am unsure what the authors mean by this statement.
5. Estimate P˜(yi|xi) with CLIP zero-shot classifier: Re-formulating the CLIP loss, including the fact that sampling at a prompt level might yield a better ZS estimate. None of this is new, but it reads as though the authors are claiming this formulation as new. It will help to state that this is a reformulation of the standard CLIP ZS process, the only addition being a different prompt template, basically just Eq. 4. In addition, multiple prompt generation uses another language model that has its own biases which are conveniently not accounted for in the main text and case into the appendix.
6. Using CLIP is suboptimal in one key manner since we dont have access to the training set, we are unsure of the biases existing in the CLIP model.
7. Section 4.2:  “synthetic symmetric/asymmetric noise.” What is this noise model?

---

> ### Author Response · Authors · 2023-11-13
> **Reply to Reviewer GWvS**
>
> Thanks for the insightful reviews and suggestions. We are sorry about the confusion caused by the presentation and we appreciate that the reviewer  checked the appendix to find some  important points-- we will be happy to move material between the main paper and the appendix as the reviewer suggests. We are happy to provide further clarifications below:
>
> >*Q1: Eq1: Sample selection mechanism takes as input the predicted probability and the label? Please clarify.*
>
> Yes, as formalized in sec 3.1, current sample selection methods usually rely on an estimated probability vector and noisy labels. Intuitively, a ‘distance’ or ‘consistency’ is calculated between the estimated probability and the annotated label, to measure the ‘cleaness’ of the label, such as the per-sample losses.
>
> >*Q2: Typo: Sec 3.2: “we consistent the notations for CLIP’s training”*
>
> We apologize for the typos here, we will fix it as ‘We make the notations consistent for CLIP's training’.
>
> >*Q3: Clarify Sec 4.1 “clean test set of in-question noisy datasets”*
>
> We apologize for any confusion. To clarify, when we mention ‘the clean test set of in-question noisy datasets’, we mean the clean test set corresponding to the noisy train set. We will address and rectify this point in the updated version of the paper.
>
> >*Q4: Appendix F: "label noise normally does not affect the sample itself": Label noise can be highly sample dependent, so I am unsure what the authors mean by this statement.*
>
> Following previous works in LNL community, we assume the causal mechanism between sample $x$ and label $y$ as $x \rightarrow y$. Thus we assume the conditional probability ($P(y|x)$ - noisy labelling) will not affect the sample coefficient prior ($P(x)$).
>
> >*Q5: Estimate P˜(yi|xi) with CLIP zero-shot classifier: Re-formulating the CLIP loss, including the fact that sampling at a prompt level might yield a better ZS estimate. None of this is new, but it reads as though the authors are claiming this formulation as new. It will help to state that this is a reformulation of the standard CLIP ZS process, the only addition being a different prompt template, basically just Eq. 4. In addition, multiple prompt generation uses another language model that has its own biases which are conveniently not accounted for in the main text and case into the appendix.*
>
> Sorry for the confusion here. We do not claim the CLIP zero-shot classification as our contribution, rather we simply want to present the multi-feature prompting in our framework. We will improve the presentation in the updated version.
>
> >*Q6: Using CLIP is suboptimal in one key manner since we dont have access to the train set, we are unsure of the biases existing in the CLIP model.*
>
> Indeed, biases, errors and domain shifts are risks inherent when using any pre-trained model. With a very large train set, we expect the CLIP model can cover a wide range of domains. In comparison to ordinary vision-only pre-trained models, CLIP still possesses its unique advantages due to its language modality and zero-shot classification capability, offering sample selection that is not influenced by noisy labels.
>
> >*Q7: Section 4.2: “synthetic symmetric/asymmetric noise.” What is this noise model?*
>
> We follow the widely-acknowledged definitions of symmetric and asymmetric noise in the LNL community. In particular, symmetric noise is generated through random label flipping, while asymmetric noise is induced by specific transitions between semantically similar classes, such as 'horse' ↔ 'deer' for the CIFAR-10 dataset.
>
> If the reviewer has any additional questions or anything else to discuss, we are more than happy to engage further in the conversation.

---

> > ### Author Response · Authors · 2023-11-22
> > **Further reply**
> >
> > Thank you very much for your time and effort in reading and improving our paper. According to your suggestions, we have made some modifications to the manuscript - we are happy to make more modifications according to your further suggestions. As the discussion period is coming to an end, we would like to know if you have any further questions. Thank you again for your time and consideration.

---

### Official Review · Reviewer_s2nG · 2023-11-09

**Soundness:** 3 good
**Presentation:** 4 excellent
**Contribution:** 3 good
**Rating:** 5
**Confidence:** 4

**Summary:**

This paper proposes a new method for learning with noisy labels, which focuses on selecting examples with vision-language models to alleviate the self-confirmation bias in vision-only models. Experiments on synthetic and real-world datasets are conducted to support the proposed method.

**Strengths:**

The idea to exploit V-L models to address the self-confirmation problem is reasonable and interesting.
The presentation is clear.

**Weaknesses:**

The second method to estimate the \tilde{p}_{y|x} seems similar to the estimation of p_{y|x} with noisy labels. Since the classifier is learned with noisy data, how can it be used to estimate the clean probability? Authors should provide more explanation for this problem.

The results are inferior to many state-of-the-art methods, such as Unicon, PES, etc.

**Questions:**

Please clarify the concerns stated in the weakness section.

---

> ### Author Response · Authors · 2023-11-13
> **Reply to Reviewer s2nG**
>
> Thanks for the insightful reviews and acknowledgement of our exploration. We are happy to answer your raised questions below:
>
> > *Q1.The second method to estimate the \tilde{p}{y|x} seems similar to the estimation of p{y|x} with noisy labels. Since the classifier is learned with noisy data, how can it be used to estimate the clean probability? Authors should provide more explanation for this problem.*
>
> We appreciate your concerns. As elucidated in Theorem 2, the second method is affected by label noise since we train a new classifier based on it. This is the motivation behind presenting two distinct methods for estimating $P(y|x)$ using CLIP. We believe that CLIP, with its unique language modality compared to ordinary pre-trained visual models, holds a significant advantage, particularly because the first method is entirely free from noise.
>
> > *Q2.The results are inferior to many state-of-the-art methods, such as Unicon, PES, etc.*
>
> We appreciate your concerns and would like to emphasize that our method is not mutually exclusive but can be used in conjunction with existing techniques. For example, we can utilize CLIPSelector seamlessly with any existing methods including the mentioned UNICON and PES, by adding an extra warmup stage with only the selected samples from CLIPSelector. Due to time limitation, as an illustration, we present results after incorporating model co-training (one of the validated effective and simple technique) with our method (by exchanging the selected samples between two models) on CIFAR100 and Animal-10N datasets (Table R1), further demonstrating that our method has great potential along with existing techniques and it is comparable with existing works.
>
> | Dataset           | CIFAR100 0.5sym | CIFAR100 0.9sym | Animal-10N |
> |-------------------|-----------------|-----------------|------------|
> | Ours              | 75.23           | 63.11           | 88.14      |
> | Ours + Co-training | 77.51           | 66.72           | 88.79      |
> | UNICON [1] | 77.6           | 44.8           | \      |
> | PES [2] | 74.3           | \           | \      |
> Table R1. Effect of co-training
>
> We will include above discussions in the updated version.If the reviewer has any additional questions or anything else to discuss, we are more than happy to engage further in the conversation.
>
> *[1] UNICON: Combating Label Noise Through Uniform Selection and Contrastive Learning, CVPR2022.*
>
> *[2] Understanding and Improving Early Stopping for Learning with Noisy Labels, NeurIPS2021.*

---

> > ### Comment · Reviewer_s2nG · 2023-11-22
> >
> > I appreciate the results in Table R1. However, the answer for Q1 is not totally convincing for me. Two distinct methods should be both reasonable by themselves.

---

> > > ### Author Response · Authors · 2023-11-22
> > > **Further clarification**
> > >
> > > We appreciate your prompt reply. We would like to further clarify your concerns. We understand your specific concerns about the estimation bias. We would like to kindly note that both methods are "reasonable", but they both have their own "limitations". The second method, in fact, is inherently affected by noisy labels like all LNL methods, as they all rely on the noisy labels to train a model. Despite, Theorem 2 (page 18 - Appendix G) provides a preliminary support that we can improve the estimation by balancing the model capacity/dataset size, etc. For example, upon CLIP vision encoder, we only obtain a linear classifier, which controls the model capacity, thereby reducing the risk of overfitting to noisy labels. Besides, due to the good representation of CLIP model, we expect even this simple linear classifier to be able to make good estimates.
> > >
> > > Thank you again for your time and consideration.

---

### Author Response · Authors · 2023-11-21
**Updating manuscript**

Dear Reviewers,


Thank you for taking the time to review our paper and for your valuable suggestions. We have made the revisions to the manuscript (Notable changes are highlighted in *blue* for your reference.):

1. We have corrected ambiguities in the presentation as well as spelling and grammatical errors.
2. We have reorganized the paper structure, moving the *hyperparameter ablation* and *additional zero-shot classification experiments* to the main paper, and moving Theorems 1 and 2 to Appendix F.
3. We have added additional experimental results in the appendix, including experiments with *fine-tuning based on the CLIP model* and with *additional model co-training*.


As the author/reviewer discussion is nearing its end, we would like to know if our response has addressed your main concerns. If so, we would kindly request that you reconsider your score. If you have any further suggestions for the paper and/or our rebuttal, please let us know. We would be happy to engage in further discussion and manuscript improvement.


Thank you again for your time and effort in improving our paper.


Sincerely,

Paper 3300 Authors

---

### Author Response · Authors · 2023-11-21
**Additional comments about the fairness and contribution of proposed method**

We would like to first express our deepest gratitude to all reviewers in advance for their invaluable time to read this 'very long' comment. :)

We surmise that the main concern of the reviewers is still the potential unfairness of the comparison with existing methods. Indeed, CLIP is clearly a powerful and influential model, and it has been applied in many other domains. Therefore, it is easy to **"misunderstand"** that "our proposed method works well simply because it is built on CLIP, and thus it is not convincing or novel".

I would like to start with a few comments on one of the most the influential work in LNL community recently - *DivideMix*. At first glance, it may seem like 'a simple combination of semi-supervised learning with a small loss mechanism'. However, it has, in fact, inspired and dominated the semi-supervision paradigm adopted by almost all recent sample selection works. *In fact, some ideas only seem natural and straightforward in hindsight.* We believe that introducing CLIP into the community of Learning with Noisy Labels(LNL) is both reasonable and meaningful. This, to some extent, is also the motivation and contribution of this paper - to motivate other researchers to explore and propose alternative strategies utilizing CLIP. Our target is never to propose a compound framework for LNL involving CLIP and beat all possible works, otherwise we will integrate existing mature techniques such as model co-training or self-supervised contrastive learning, instead of current simple training scheme.

**In fact, we are glad that several reviewers pose questions such as `'How would replacing the encoder of existing methods with CLIP's encoder impact the results?'` - this is exactly the insight we expect of this paper for the community**. However, as emphasized, this is beyond the scope of this work and discussed in the paper already (*page 6 - To fully explore CLIP?*). The reason we did not adopt this style is also because CLIP's role in LNL work is still underexplored - Without an acknowledged evaluation protocol (the pretrained CLIP encoder has fixed model size and capacity different with common acknowledged baselines), it is hard to evaluate such methods comparing with existing methods - We are also confident that as CLIP's adoption in the LNL domain expands, standardized benchmark protocols will emerge, avoiding such challenges faced by future researchers. To this end, to ensure a fair and efficient comparison currently, we opted for offline sample selection. Specifically,  because:

- **Acknowledgement: sample selection is the current dominant paradigm for LNL, by utilizing CLIP for sample selection our method can be seamlessly utilized with existing methods.**

- **Effectiveness: Viewing CLIP as a 'black-box foundation model', considering the possible computational cost/deployment issues, etc, the most direct and convenient way to utilize it would be the offline one.**

Another potential fair comparison involving existing methods is to solely utilize our proposed sample selection module - CLIPSelector, as a plug-in module for existing methods and ablate the effect of CLIP. Please note, similarly, such experiments are still to show the benefits of CLIP, which has already been validated by our experiments, rather than to discriminate between exsiting methods or propose new methods.

---

### Meta-Review · Area_Chair_f6ew · 2023-12-17

**Metareview:**

This paper addresses learning with noisy labels, and proposes to use a large-scale vision-language model (CLIP) for sample selection in order to mitigate a self-confirmation bias in vision models.

While the reviewers acknowledged the importance of this study, they raised several concerns that were viewed by AC as the critical issues:
(1) presentation clarity and rigour – see detailed comments of the Reviewer GWvS on how to improve;
(2) insufficient experiments and baseline comparisons to assess the efficacy of the proposed approach – see Reviewer tEQg extensive comments during discussions, including a need for an ablation study on the replacement of the baselines' backbone with CLIP's visual encoder – see Reviewer tEQg and Reviewer Vnif comments; lack of baseline comparisons – see Reviewer s2nG comments.

The rebuttal was able to clarify some questions, but did not manage to sway any of the reviewers. In light of an unanimous lack of enthusiasm for this work, a general consensus among reviewers and AC was reached to reject the paper. Introducing CLIP into the community of Learning with Noisy Labels(LNL) would be most valuable and appreciated at a dedicated venue, e.g. a workshop. We hope the reviews are useful for improving and revising the paper.

**Justification For Why Not Higher Score:**

All reviewers are in consensus.

**Justification For Why Not Lower Score:**

N/A

---

### Decision · Program_Chairs · 2024-01-16

Reject